# [Re] Improving Interpretation Faithfulness for Vision Transformers

**Izabela Kurek**[*]  *izabela.kurek@student.uva.nl*
*University of Amsterdam*

**Wojciech Trejter**[*]  *wojciech.trejter@student.uva.nl*
*University of Amsterdam*

**Stipe Frković**[*]  *stipe.frkovic@student.uva.nl*
*University of Amsterdam*

**Andro Erdelez**[*]  *andro.erdelez@student.uva.nl*
*University of Amsterdam*

**Reviewed on OpenReview:** *https://openreview.net/forum?id=ZODhgU8fBt*

## Abstract

This work aims to reproduce the results of Faithful Vision Transformers (FViTs) proposed by Hu et al. (2024) alongside interpretability methods for Vision Transformers from Chefer et al. (2021) and Xu et al. (2022). We investigate claims made by Hu et al. (2024), namely that the usage of Diffusion Denoised Smoothing (DDS) improves interpretability robustness to (1) attacks in a segmentation task and (2) perturbation and attacks in a classification task. We also extend the original study by investigating the authors' claims that adding DDS to any interpretability method can improve its robustness under attack. This is tested on baseline methods and the recently proposed Attribution Rollout method. In addition, we measure the computational costs and environmental impact of obtaining an FViT through DDS. Our results broadly agree with the original study's findings, although minor discrepancies were found and discussed.

## 1 Introduction

By adopting attention mechanisms that constitute transformers (Vaswani et al., 2023), the field of computer vision has experienced significant breakthroughs in tasks such as image recognition (Dosovitskiy et al., 2021), object detection (Zhu et al., 2021), image processing (Chen et al., 2021), and semantic segmentation (Zheng et al., 2021). A notable success is the Vision Transformer (ViT) proposed by Dosovitskiy et al. (2021), which, in addition to its state-of-the-art performance, provides an attention vector that can be used to explain its decision-making. However, this approach is not robust to slight input perturbations, leading not only to false classification but also to attention vectors that are misleading in interpretability.

Traditional post-hoc interpretability methods have been widely used to analyse ViTs by highlighting important regions in an input image based on attention or gradient-based explanations. They obtain the intermediate self-attention matrices of the model for an input and aggregate them alongside heads and layers. Those aggregations can happen using various components of the model. Raw Attention (Vaswani et al., 2023) directly uses the attention scores from the last self-attention layer to determine importance (Vaswani et al., 2023). GradCAM (Selvaraju et al., 2019) uses gradient information obtained during backpropagation to determine the importance of each pixel (Selvaraju et al., 2019). Rollout (Abnar & Zuidema, 2020) assumes that all attention heads contribute equally and averages them to estimate importance (Abnar & Zuidema,

---

[*]Equal contributions.

2020). Layer-wise Relevance Propagation (LRP) propagates relevance scores backwards through the network to estimate pixel importances (Binder et al., 2016). Finally, Transformer Attribution (TA) integrates both gradients and relevance scores to find important pixels (Chefer et al., 2021). The methods output relevance scores for each pixel of the input image, which can be converted into an attention heat map to visualize how relevance regions.

Despite the progress of interpretability methods, the vulnerability to adversarial attacks remains a critical limitation, as small input perturbations can drastically alter their explanations, undermining their reliability. Hu et al. (2024) address this issue by proposing a Faithful ViT (FViT), a vision transformer that is robust to both performance and interpretability concerns. Unlike standard ViTs, where adversarial perturbations significantly affect interpretability, FViTs ensure that explanations remain stable even under such attacks.

To convert a ViT into an FViT, they propose Denoised Diffusion Smoothing (DDS), a novel plug-in method that combines randomized smoothing and diffusion-based denoising. Specifically, it adds Gaussian noise to an input sample, which is then processed through a guided diffusion model to remove the noise. The resulting samples are then fed into a ViT of choice for further analysis.

FViTs also contribute to the ongoing debate between post-hoc and model-based interpretability. While post-hoc methods analyze pre-trained models after training to explain predictions, they often lack robustness to perturbations. In contrast, FViTs integrate interpretability directly into the model, promising that explanations remain stable under adversarial attacks, thus improving both task performance and model faithfulness.

This paper focuses on reproducing Hu et al. (2024)'s experiments, extending the original work by investigating and applying DDS in novel ways, and evaluating the soundness of their main claims. Our results corroborate Hu et al. (2024)'s findings that DDS enhances model interpretability and robustness against adversarial attacks, thus reinforcing the broader claim that DDS is a viable method for transforming standard ViTs into interpretable and robust FViTs.

## 2 Scope of reproducibility

Hu et al. (2024) claim that (A) applying DDS to any ViT transforms it into an FViT, and that (B) FViTs, unlike standard ViTs, are robust against adversarial attacks both in task utility and, more importantly, in model interpretability. To support their claims, the authors provide mathematical proofs and conduct a series of experiments. Our paper focuses on evaluating the experiments and the main claims derived from them. The following claims are tested:

- **Claim 1:** Applying DDS to a ViT results in more robust segmentations (obtained from an interpretability method), particularly under adversarial attacks in an image segmentation task.

- **Claim 2:** Applying DDS to a ViT increases the model's classification robustness under segmentation-informed input perturbation and adversarial attack in an image classification task.

Furthermore, this paper also expands on the original work by using DDS with methods other than the original TA (Hu et al., 2024). Attribution Rollout (AR) (Xu et al., 2022), a new interpretability method, was shown to offer several advantages over TA (Chefer et al., 2021), and since no study examined DDS with AR, this paper aims to fill that gap. Furthermore, Hu et al. (2024) claim that DDS is a plug-in method to improve performance and robustness against adversarial attacks. Therefore, the addition of DDS to the baseline interpretability methods was also investigated. Namely, our extensions are:

- **Extension 1:** Investigating whether DDS with Attribution Rollout produces better interpretability results than DDS with Transformer Attribution in both aforementioned tasks.

- **Extension 2:** Investigating whether DDS is a plug-in method for improving interpretability robustness by combining it with baseline interpretability methods in an image segmentation task under adversarial attack.

Lastly, this paper brings to light the possible environmental impacts of applying DDS to ViTs. An investigation of the energy usage of DDS compared to other methods was missing in the original study and was therefore performed. Hence, our study also reports these findings:

- **Impact 1:** FViT's environmental impact is significantly higher than that of all other methods.

## 3 Background

**Faithfulness**
Faithfulness refers to how accurately an explanation represents the true reasoning process of a model when making a decision (Jacovi & Goldberg, 2020). It is based on three distinctive assumptions:

- **The Model Assumption:** *Two models will make the same prediction if and only if they use the same reasoning process.*

- **The Prediction Assumption:** *On similar inputs, the model makes similar decisions if and only if its reasoning is similar.*

- **The Linearity Assumption:** *Certain input parts are more important to the model reasoning than others. Moreover, the contributions of different parts of the input are independent of each other.*

**Robustness**
Model robustness denotes the capacity of a model to sustain stable predictive performance in the face of variations and changes in the input data (Braiek & Khomh, 2024). Specifically, this paper focuses on adversarial robustness, which is the extent to which models resist adversarial perturbations.

**DDS**
DDS is a method for transforming standard ViTs into FViTs with improved interpretability and robustness. It combines randomized smoothing using Gaussian noise with a denoising diffusion probabilistic model, enhancing the faithfulness of attention maps. Gaussian noise is shown to be nearly optimal for this purpose under both $\ell_2$ and $\ell_\infty$ norms.

**Interpretability methods**
To assess faithfulness, this paper uses post-hoc interpretability methods that produce heat maps highlighting which parts of an image influence the model's decision. These include:

- **Raw Attention:** Uses attention weights from the final layer.

- **GradCAM:** Highlights important regions using gradient-based signals.

- **Rollout:** Aggregates attention across layers to show how information flows.

- **LRP:** Propagates output relevance back through the model.

- **Transformer Attribution:** Combines gradients and relevance for detailed maps.

## 4 Methodology

Two experiments were carried out to investigate the two aforementioned claims: an image segmentation task under a projected gradient descent (PGD) adversarial attack, and an image classification task under positive and negative perturbation as well as a PGD attack, as described by Hu et al. (2024) in addition to Chefer et al. (2021). Pre-trained ViT and DeiT models were used for the segmentation task, whereas a pre-trained ViT was used for the classification task (Dosovitskiy et al., 2021; Touvron et al., 2021). Interpretability methods used in Hu et al. (2024); Chefer et al. (2021) as well as Attribution Rollout by Xu et al. (2022) were investigated. A model combined with an interpretability method produces attention heat maps, which are

used to create image segmentations in the segmentation task as well as inform the regions for the perturbation attacks in the classification task.

Furthermore, the original study implemented a qualitative test of the interpretability methods under an attack; attention heat maps computed by the interpretability methods were provided before and after the attack. We recreated these results following the provided demo implementation.

Moreover, we extend the study by investigating the plug-in claims of DDS as well as the potential improvements of AR. Consequently, the addition of DDS to baseline methods was investigated in the image segmentation task, while the addition to AR was investigated in both segmentation and classification tasks as described. New visualizations are also provided.

The energy usage reported by the cluster was also measured, which was combined with publicly available estimates on national grid carbon efficiency, to estimate the environmental impact.

The code used for the experiments was built on top of Hu et al. (2024), which extended the code from Chefer et al. (2021) by providing the qualitative demo. We further extended Chefer et al. (2021)'s code by adding PGD and DDS to both tasks based on the demo implementation by Hu et al. (2024). We also added the AR interpretability method to the segmentation task following the implementation by Xu et al. (2022). Our code is available at github.com/aerdelez/re-fvit.

### 4.1 Model and Algorithm Descriptions

Following the original study, a ViT-Base model introduced by Dosovitskiy et al. (2021) was used for both tasks. It was pre-trained on the ImageNet-21k dataset and uses a resolution of 224x224.[1] Similarly, a DeiT-base model introduced by Touvron et al. (2021) was also used for the segmentation task.[2] Moreover, a class-unconditional ImageNet diffusion model at a resolution of 256x256 (Dhariwal & Nichol, 2021) was used as the diffusion model in the DDS method.[3]

In the Transformer architecture, interpretability methods obtain the intermediate self-attention matrices of the model for an input and aggregate them alongside heads and layers using various components of the model. The pixel relevance scores are used to create attention heat maps of the input image. All interpretability methods included in Hu et al. (2024) (baselines) are used in the study; these include Raw Attention, GradCAM, Rollout, LRP and TA. The baseline GradCAM and LRP methods were not investigated in the classification task due to long experiment runtimes and ordinary performance in the original study as well as the reproduced segmentation task. As mentioned, AR was also investigated and implemented according to (Xu et al., 2022). Default hyperparameters were used for all of the methods.

The DDS method was implemented following the algorithm description included in the Appendix of Hu et al. (2024). That is, to convert a ViT into an FViT, Gaussian noise is added to the input sample. That sample is then denoised using an external model (in our case, a guided diffusion model). After this process, the de-blurred sample is passed to a ViT of choice.

### 4.2 Metrics

In the scope of the reproduction as well as in the extensions, we make use of several performance metrics, which are defined as follows.

**Pixel Accuracy** (Pix. Acc.) Defined as the proportion of all true positive predictions $TP$ over the number of pixels in the image $n$, i.e. $\frac{TP}{n}$.
**Mean Intersection over Union** (mIoU) Defined as the mean of the ratios of intersection and union for each of the $n_{class}$ classes, i.e. $\frac{1}{n_{class}} \sum_{i=1}^{n_{class}} \text{IoU}_i$. In our experiments, we use $n_{class} = 2$ with the classes representing the foreground and background.
**Mean Average Precision** (mAP) Defined as the mean average precision of the segmentation.

---

[1]Obtained from `https://github.com/huggingface/pytorch-image-models/releases/download/v0.1-vitjx/jx_vit_base_p16_224-80ecf9dd.pth`
[2]Obtained from `https://dl.fbaipublicfiles.com/deit/deit_base_patch16_224-b5f2ef4d.pth`
[3]Obtained from `https://github.com/openai/guided-diffusion`

### 4.3 Datasets

The ImageNet-segmentation subset was used for the image segmentation task (Guillaumin et al., 2014).[4] The subset contains images sourced from the full ImageNet dataset with ground-truth segmentations. Therefore, with a model pre-trained for ImageNet classification, there is no need for fine-tuning or data splitting. The subset consists of 4276 images encompassing 445 classes and 6225 segmentations. The segmentations are binary and only distinguish the foreground from the background. COCO (Lin et al., 2014) and Cityscape (Cordts et al., 2016) datasets were not used due to a lack of experiment scripts, pre-trained model weights, and computational resources needed to train the ViT and diffusion models.

The ImageNet LSVRC 2012 Validation Set (Russakovsky et al., 2015) was used for the classification under perturbation and attack task.[5] The set contains 50,000 images with corresponding labels equally distributed from 50 classes. Generating attention heat maps for the full set of images was not feasible due to both computational and time constraints. Therefore, sampling with equal probability weights, no replacement and a preset seed was used. A total of 4000 images were sampled from the validation dataset. This means that the results are not fully representative but should still indicate the general trends of the algorithms.

### 4.4 Hyperparameters

No hyperparameter search was performed for the experiments; pre-trained ViT and diffusion models were used. Unless specified otherwise, the replicated experiments use the default hyperparameters proposed for DDS, which are a noise level of $\frac{8}{255}$ and 45 backward steps for denoising. The process of finding the faithfulness region of an FViT extends the typical process found in explainable ViTs by taking the average of multiple samples and performing additional computations. Following the demo implementation of Hu et al. (2024), the number of samples created was 10 for the qualitative visualizations before adversarial attack and 2 in all other cases.

A PGD attack was used in both tasks; the parameters were left to their default implementation values, as used by Hu et al. (2024). Specifically, unless specified otherwise, the maximum perturbation was set to $\frac{8}{255}$, the step size to $\frac{2}{255}$, and the number of steps to 10. PGD implementation comes from `torchattacks` library, which is based on the $\ell_\infty$ norm.

### 4.5 Experimental Setup

Following Hu et al. (2024), Chefer et al. (2021), and Xu et al. (2022), in the image segmentation task, the visualization is considered as a soft-segmentation of the (PGD-attacked) image and compared to the ImageNet ground truth segmentation. Performance is measured by pixel accuracy, obtained after thresholding each visualization by the mean value, mIoU, and mAP, which uses a soft-segmentation to obtain a score that is threshold-invariant.

Again, following Hu et al. (2024) and Chefer et al. (2021), the image classification under perturbation and attack tests follow a two-stage setting. First, a PGD attack is applied or not applied, depending on the experiment. Then, a model with an interpretability method (with or without DDS) is used to extract the visualizations from the ImageNet dataset. Then, pixels are masked from an input image and the mean top-1 accuracy of the model is measured. In positive perturbation, pixels are masked from the highest relevance to the lowest, while in negative perturbation from the lowest to the highest. In positive perturbation, a decrease in performance is expected, while in negative perturbation, accuracy should be maintained. Both tests used perturbation in the range of 10 to 90% of the pixels in 10% increments. The AUC of top-1 accuracy over perturbation steps was calculated.

A random seed of 44 was used for all experiments.

---

[4]Obtained from `https://calvin-vision.net/bigstuff/proj-imagenet/data/gtsegs_ijcv.mat`
[5]Obtained from `https://academictorrents.com/details/5d6d0df7ed81efd49ca99ea4737e0ae5e3a5f2e5` and `https://image-net.org/data/ILSVRC/2012/ILSVRC2012_devkit_t12.tar.gz`

### 4.6 Computational Requirements

The experiments were run on an NVIDIA A100 GPU partitioned into two instances using Multi-Instance GPU technology, effectively utilizing half of the GPU; in addition, 9 cores of Intel Xeon CPUs and 60GB of RAM were utilized. Table 4 in Appendix A shows the computational resources used in the segmentation and classification tasks.

## 5 Mathematical Foundation

In the original paper, the authors present a formal mathematical definition of what constitutes a Faithful Vision Transformer and a proof that the proposed method of Denoised Diffusion Smoothing indeed results in an FViT agreeing with the aforementioned definition. In this section, these definitions and theorems will be discussed with the aim of this paper not only to provide a reproduction of experimental results, but also to evaluate the proposed mathematical framework of FViTs.

The definitions and theorems presented follow closely the ones put forward by Hu et al. (2024), but have been slightly rephrased to improve readability. Additionally, our full replications of the authors' proofs can be found in the Appendix.

### 5.1 Faithful ViTs

Let us begin by discussing the definition of a Faithful Vision Transformer itself.

**Definition 5.1.** *We call a function $f : \mathbb{R}^{q \times n} \to \mathbb{R}^n$ an $(R, D, \gamma, \beta, k, \|\cdot\|)$-faithful attention module for ViTs if for any given input data $x$ and for all $x' \in \mathbb{R}^{q \times n}$ such that $\|x - x'\| \leqslant R$, $f(x')$ satisfies*

1. *(Top-k robustness) $V_k(f(x'), f(x)) \geqslant \beta$*

2. *(Prediction robustness) $D(\bar{y}(x'), \bar{y}(x)) \leqslant \gamma$, where $\bar{y}(x'), \bar{y}(x)$ are the prediction distributions of ViTs based on $f(x'), f(x)$ respectively*

*where $V_k(x, x') = \frac{1}{k}|T_k(x) \cap T_k(x')|$ is a top-k overlap ratio of two vectors. We also call the vector $f(x)$ an $(R, D, \gamma, \beta, k, \|\cdot\|)$-faithful attention for $x$ and the ViTs based on $f$ as faithful ViTs (FViTs).*

While Definition 5.1 presents a valuable formal framework of defining faithfulness in Vision Transformers, it seems to go against the broadly accepted definition of faithfulness in artificial intelligence, which usually consists of three criteria: Proximity, Connectedness, and Stability (Laugel et al., 2019). In contrast, this definition seems to emphasize stability alone with its priority on robustness criteria.

For its utility in further proofs, we also introduce the Rényi Divergence.

**Definition 5.2.** *Given two probability distributions $P$ and $Q$ and $\alpha \in (1, \infty)$, the $\alpha$-Rényi divergence $D_\alpha(P\|Q)$ is defined as*

$$D_\alpha(P\|Q) = \frac{1}{1-\alpha} \log \mathbb{E}_{x \sim Q} \left( \frac{P(x)}{Q(x)} \right)^\alpha.$$

### 5.2 Denoised Diffusion Smoothing and Faithfulness

After introducing necessary definitions for FViTs, the authors of the original paper put forward a number of theorems allowing us to determine whether a ViT is faithful under specific conditions. In this paper, only the $\ell_2$ is discussed as all the analogous theorems and their proofs are of high similarity. Of the two theorems approached, we were not able to verify the correctness in the form they were presented for either of them. In this section, we will present revised versions of the theorems, while their original wording (referred to as Conjectures) can be found in Appendix B.

Authors do not present a proof for Conjecture B.1 in their paper and it was found that there exists a case where this statement presented as a theorem is not true. A revised version of this statement is the following theorem:

**Theorem 5.1.** *If a function is an $(R, D_\alpha, \gamma, \beta, k, \|\cdot\|)$-faithful attention module for ViTs then if*

$$\gamma < -\log\left((1 - p_{(1)} - p_{(2)} + 2\left(\frac{1}{2}(p_{(1)}^{1-\alpha} + p_{(2)}^{1-\alpha})\right)^{\frac{1}{1-\alpha}}\right),$$

*we have for all $x'$ such that where $\|x - x'\| \leqslant R$,*

$$\arg\max_{g \in \mathcal{G}} \mathbb{P}(\bar{y}(x) = g) = \arg\max_{g \in \mathcal{G}} \mathbb{P}(\bar{y}(x') = g)$$

*where $\mathcal{G}$ is the set of classes, $p_{(i)}$ are i-th largest probabilities in $\{p_j\}$, where $p_j$ is the probability that $\bar{y}(x)$ returns the j-th class.*

In short, Theorem 5.1 allows us to make a guarantee that for a faithful attention module in a discrete classification scenario, within a certain region, the final prediction of the classifier will not be affected by an alteration of the input data $x$. A proof for Theorem 5.1 and at the same time a counterexample to Conjecture B.1 will be shown in Appendix B.1.

We believe the proof presented in Hu et al. (2024) for Conjecture B.2 is not sufficient to conclude the truthfulness of the presented statement. Authors of the original paper make use of the term *utility robustness* in their proof, in fact making the second of the two terms in the max function of the faithfulness criterion in the theorem based on utility robustness criterion[6]. Utility robustness, however, is neither defined nor mentioned anywhere else in the original paper and it is not stated as one of the criteria for an attention module to be considered faithful (as per Definition 5.1). Therefore, its use as a criterion for faithfulness in the original proof of Theorem 5.2 is quite perplexing. Additionally, in their proof, the authors do present a required upper bound to guarantee prediction robustness, but they chose to forego its inclusion in Conjecture B.2.

Looking at the first element in the maximum, we were not able to follow the proof for that sufficiently well to include it in the revised version of the theorem. Instead, another proof has been presented using a result from Liu et al. (2021), which Hu et al. (2024) cite as inspiration for their proof. However, in Hu et al. (2024)'s version of the proof (specifically Lemma B.2) they make choices which we found confusing. They aim to find $\min_{q \in \mathcal{Q}, V_k(\tilde{w}, q) \geqslant \beta} D_\alpha(\tilde{w}\|q)$ (minimum divergence to *keep* top-k robustness) and then upper bound Rényi divergence with that result to guarantee top-k robustness. In our understanding, we should be finding the minimum divergence to *violate* top-k robustness then use that as an upper bound, as then we guarantee that a violation of top-k robustness is impossible. These are the steps taken by Liu et al. (2021). Thus, we chose to present this result instead.

A revised version of the theorem, including the upper bound necessary for guaranteeing prediction robustness, is found below.

**Theorem 5.2.** *Consider the function $\tilde{w}$ where $\tilde{w} = Z(T(x + z))$, with $Z$ being the self attention module, $T$ a denoised diffusion model, $x$ input data and $z \sim \mathcal{N}(0, \sigma^2 I_{q \times n})$. Then it is an $(R, D_\alpha, \gamma, \beta, k, \|\cdot\|)$-faithful attention module for ViTs for any $\alpha > 1$ if for any input data $x$ we have*

$$\sigma^2 \leqslant \max\left\{\frac{\alpha R^2}{-2\log\left(2k_0\left(\frac{1}{2k_0}\sum_{i \in \mathcal{S}}\tilde{w}_{i^*}^{1-\alpha}\right)^{\frac{1}{1-\alpha}} + \sum_{i \notin \mathcal{S}}\tilde{w}_{i^*}\right)},\right.$$

$$\left.\frac{\alpha R^2}{-2\log\left(1 - p_{(1)} - p_{(2)} + 2\left(\frac{1}{2}\left(p_{(1)}^{1-\alpha} + p_{(2)}^{1-\alpha}\right)\right)^{\frac{1}{1-\alpha}}\right)}\right\}$$

*where $k_0 = \lfloor(1-\beta)k\rfloor + 1$ and $\mathcal{S}$ the set of last $k_0$ components in top-k indices and the top $k_0$ components out of top-k indices.*

Theorem 5.2 allows us to make a guarantee that an attention module (and henceforth a ViT using it) is faithful under specific conditions. The proof is available in the Appendix B.2.

---

[6]More detail in Appendix B.2

### 5.3 Algorithms

In their paper, Hu et al. (2024) present two algorithms based on the previously presented mathematical framework, allowing for finding FViTs computationally. Taking into account aforementioned issues, we cannot guarantee the correctness of Algorithm 2's ability to find the faithfulness region of an FViT, as it is based on Conjecture B.2. Additionally, Algorithm 1 of Hu et al. (2024) is an adapted Algorithm 1 put forward by Carlini et al. (2023) for guaranteeing adversarial robustness of any classifier. Taking into account our previous note that Definition 5.1 is more similar to that of robustness than faithfulness, while we can make guarantees based on Definition 5.1 (based on revised Theorems 5.1 and 5.2), we would rather call it a robust ViT.

## 6 Experimental Results

Briefly, our results in both the segmentation and classification tasks exhibit the same general trends as in Hu et al. (2024), thereby confirming Claims 1 and 2 in Section 2. Namely, explanations obtained from an FViT are indeed more robust under attack compared to regular ViT. However, baseline methods perform worse than in the original study. Instead, the obtained metrics are more consistent with those from Chefer et al. (2021).

Applying DDS to Attribution Rollout substantially improved its performance and also achieved the best overall results in both image segmentation and classification. Applying DDS to the baseline interpretability methods, in image segmentation, saw an improvement only for LRP, with no change for the other methods, thereby somewhat confirming the plug-in nature of DDS. Lastly, applying DDS to a ViT resulted in a substantial adverse increase in the environmental impact.

### 6.1 Reproducing the Original Study

#### 6.1.1 Image Segmentation

Table 1 shows the reproduced and original results of interpretability methods investigated by Hu et al. (2024) in the image segmentation task under a PGD attack.

| Method | Pix. Acc. | | mIoU | | mAP | |
|---|---|---|---|---|---|---|
| | Ours | Hu et al. | Ours | Hu et al. | Ours | Hu et al. |
| Raw Attention | 0.64 | 0.65 | 0.43 | 0.54 | 0.78 | 0.82 |
| Rollout | 0.72 | 0.67 | 0.52 | 0.56 | 0.82 | 0.84 |
| GradCAM | 0.67 | 0.69 | 0.42 | 0.58 | 0.72 | 0.86 |
| LRP | 0.66 | 0.71 | 0.40 | 0.6 | 0.64 | 0.88 |
| TA | 0.73 | 0.73 | 0.52 | 0.62 | 0.82 | 0.9 |
| TA + DDS | **0.77** | **0.76** | **0.58** | **0.65** | **0.85** | **0.93** |

Table 1: Reproduced and original pixel accuracy, mIoU, and mAP results from the ImageNet segmentation task under a PGD attack.

Same as Hu et al. (2024), the addition of DDS improved TA over all the metrics; however, we found our baselines often did not match the results presented in the original study. For example, GradCAM results for mIoU range from 0.42 (ours) to 0.58 (Hu et al., 2024) and for mAP from 0.72 (ours) to 0.86 (Hu et al., 2024). In fact, for each of the methods, at least one of the metrics in our replication is worse compared to the original study.

Furthermore, the ordering in Hu et al. (2024) is unanimously improving: each method improved over worse ones in all metrics in steady increments. However, our results across methods were less distinct: for example, while TA had a higher pixel accuracy and mAP than Rollout, it had a lower mIoU.

Lastly, similar to Hu et al. (2024), our study obtained higher metric values with some methods under attack than Chefer et al. (2021) without attack. However, the increase was substantially smaller than that of Hu

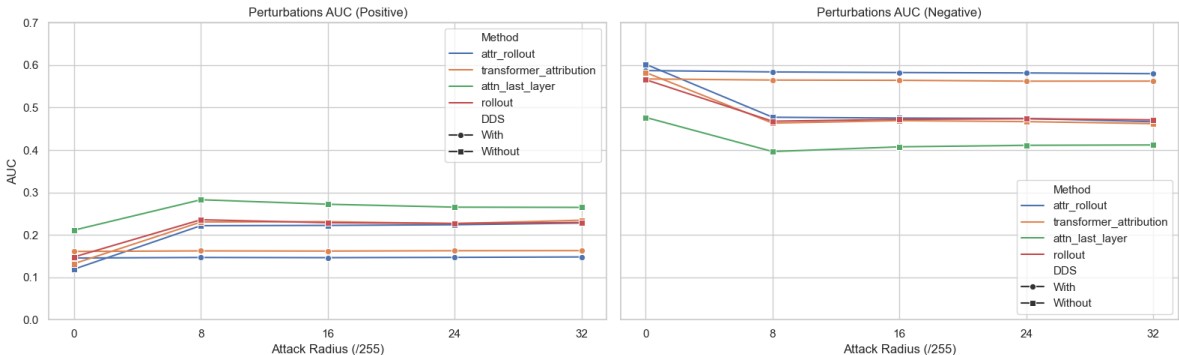

Figure 1: The AUC values of the interpretability methods in the image classification task per PGD attack radius under positive or negative perturbation. The AUC of the correctly classified images is calculated for all perturbation amounts: from 10% to 90% of the pixels removed in 10% increments. For positive perturbation, a lower AUC is better, while for negative a higher AUC is better.

et al. (2024). A comparison of some metrics obtained for TA and GradCAM across the two studies and ours can be seen in Table 2. A detailed analysis of the discrepancies in the results of Hu et al. (2024), Chefer et al. (2021), and ours is presented in Appendix C.

| Method | Attack | Pix. Acc. | mIoU | mAP |
|---|---|---|---|---|
| TA | None (Chefer et al., 2021) | **0.8** | **0.62** | 0.86 |
| | PGD (Hu et al., 2024) | 0.73 | 0.62 | ***0.9*** |
| | PGD (ours) | 0.73 | 0.52 | 0.82 |
| GradCAM | None (Chefer et al., 2021) | 0.64 | 0.41 | 0.72 |
| | PGD (Hu et al., 2024) | ***0.69*** | ***0.58*** | ***0.86*** |
| | PGD (ours) | *0.67* | *0.42* | 0.72 |

Table 2: Results from the ImageNet segmentation task for two methods from Chefer et al. (2021), Hu et al. (2024), and our study. Bold values represent the best metrics for a method across the studies, while italicized represent an increase in a metric after an attack.

### 6.1.2 Image Classification

Figure 1 shows the classification AUC for all perturbation levels for both positive and negative perturbations and different PGD attack radii.

As mentioned before, the results are on a portion of the original dataset and are therefore not completely comparable with those of Hu et al. (2024). Compared to Hu et al. (2024), attack radius values had a less significant impact on performance and the values do not match exactly. Nonetheless, the ordering of the methods and general trends of the metrics align with Hu et al.'s results, which can be seen in Figure 5 in Appendix D.

### 6.1.3 Visualizations

Figure 2 was taken from the original study (Hu et al., 2024) and visualizes the attention heat maps under adversarial corruption. The maps were reproduced by rerunning the authors' qualitative demo with the original hyperparameters and can be observed in Figure 3; as can be seen, the images are very similar to nearly identical and further show that DDS improves interpretability robustness. Further visualizations of interpretability methods, with different hyperparameters, can be found in Appendix E.

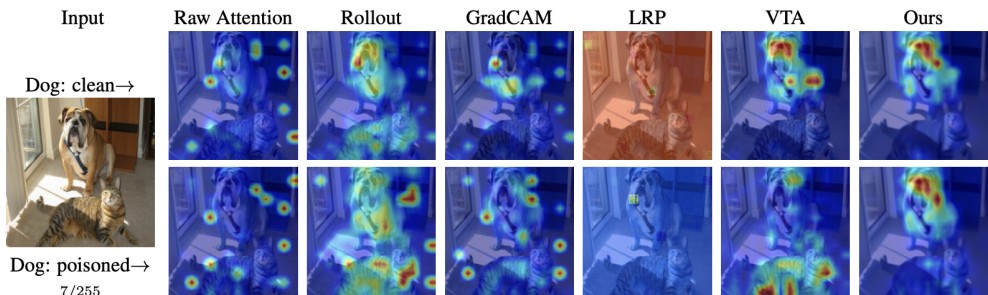

Figure 2: Attention heat maps from the original study.

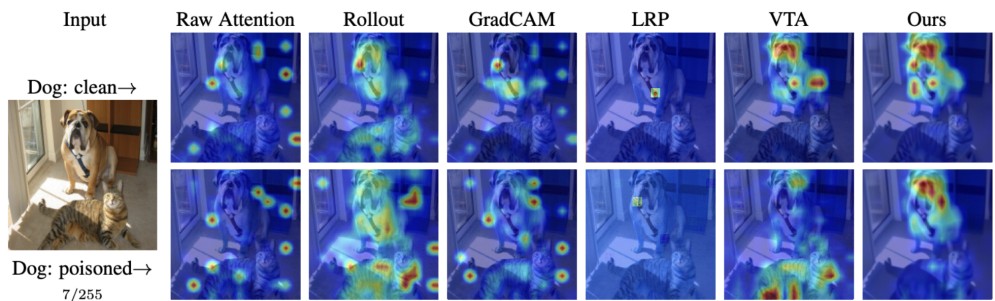

Figure 3: Reproduced attention heat maps.

## 6.2 Extending DDS With Other Methods

The results of applying DDS to the baseline interpretability methods and AR for image segmentation are shown in Table 3. As hypothesized in Section 2, adding DDS improved the performance of AR and also achieved the best overall results in image segmentation. With regards to the general plug-in nature of DDS for improving robustness, DDS resulted in similar or improved metrics across all methods and models. Specifically, an improvement was not found only with Raw Attention, Rollout, and LRP with a DeiT model.

Similarly, as hypothesized in Section 2, adding DDS improved the performance of AR and also achieved the best overall results in image classification; this can be seen in Figure 1.

## 6.3 Environmental Impact

One concern that was observed during our preliminary experiments was the computational cost of the FViT achieved through DDS. Therefore, we found it appropriate to measure the environmental impact of each interpretability method by reporting its runtime and converting it using the tool presented by Lacoste et al. (2019) with the average carbon efficiency in the Netherlands for 2024 of 370 g eqCO$_2$/kWh to validate Impact 1[7]. The results are shown in Figure 4 with full data available in Table 8 in Appendix F. In short, applying DDS had a strongly negative impact: applying it to each of the interpretability methods increased the runtime and emissions over 10 times in some cases.

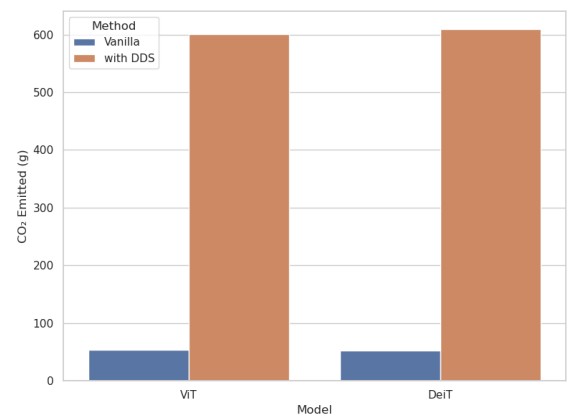

Figure 4: Mean grams of CO$_2$ emitted across interpretability methods in the image segmentation task.

---

[7]Obtained with *nowtricity* in January of 2025.

| Model | Method | DDS | Pix. Acc. | mIoU | mAP |
|-------|--------|-----|-----------|------|-----|
| ViT | Raw Attention | × | 0.64 | 0.43 | 0.78 |
|  |  | ✓ | *0.67* | *0.46* | *0.79* |
|  | Rollout | × | 0.72 | 0.52 | 0.82 |
|  |  | ✓ | *0.76* | *0.58* | *0.85* |
|  | GradCAM | × | 0.67 | 0.42 | 0.72 |
|  |  | ✓ | *0.69* | *0.47* | *0.76* |
|  | LRP | × | 0.66 | 0.40 | 0.64 |
|  |  | ✓ | 0.66 | *0.41* | *0.65* |
|  | TA | × | 0.73 | 0.52 | 0.82 |
|  |  | ✓ | *0.77* | *0.58* | *0.85* |
|  | AR | × | 0.74 | 0.53 | 0.82 |
|  |  | ✓ | **0.78** | **0.60** | **0.86** |
| DeiT | Raw Attention | × | 0.66 | 0.44 | 0.78 |
|  |  | ✓ | 0.60 | 0.38 | 0.76 |
|  | Rollout | × | 0.68 | 0.48 | 0.80 |
|  |  | ✓ | 0.67 | 0.47 | 0.80 |
|  | GradCAM | × | 0.66 | 0.39 | 0.66 |
|  |  | ✓ | *0.67* | *0.47* | *0.79* |
|  | LRP | × | 0.62 | 0.37 | 0.64 |
|  |  | ✓ | 0.62 | 0.36 | 0.64 |
|  | TA | × | 0.68 | 0.46 | 0.79 |
|  |  | ✓ | **0.79** | *0.60* | **0.85** |
|  | AR | × | 0.68 | 0.45 | 0.78 |
|  |  | ✓ | **0.79** | **0.61** | **0.85** |

Table 3: Extended pixel accuracy, mIoU, and mAP results from the ImageNet segmentation task under a PGD attack. Italicized values represent improved metrics, while bold ones represent best overall score across methods per model.

## 7 Discussion

The results confirm that applying DDS produces more robust and stable explanations across multiple tasks, interpretability methods, and ViT models. Specifically, we have confirmed that a PGD attack on the input sample is partially resolved with an FViT obtained with DDS. Unfortunately, improvements were not found across all settings. Specifically, no improvements were found for DeiT with Raw Attention, Rollout, and LRP, while very minor improvements were found for ViT with Raw Attention and LRP. A possible explanation for this is that the aforementioned methods do not utilize gradient information as opposed to methods uniformly improving under DDS.

Furthermore, we do not agree with additional claims from the authors, namely that DDS is efficient and promising for real-world applications with large-scale data. With runtime multiplied several times, it is hard to argue for the efficiency of the proposed method, which increases the pixel accuracy by a few percent under PGD attack and optimal conditions.

In our reproductions of experiments conducted by Hu et al. (2024) we observed result trends similar to Hu et al. (2024), thereby affirming the effectiveness of DDS in improving a ViT's robustness against perturbations. However, most of our experiments resulted in worse overall metrics than the original paper. We believe this discrepancy with the original paper is due to us not having access to the most up-to-date implementation by Hu et al. (2024), the existence of which we were informed of through e-mail communication with the authors, but were not provided with. As our results match more closely with Chefer et al. (2021), we believe this more recent codebase of Hu et al. (2024) might have made overall improvements to all the methods, allowing them to report these state-of-the-art results.

We and Hu et al. (2024) report higher values for GradCAM under attack than Chefer et al. (2021) without an attack (see Table 2). We double-checked implementations and all three are identical. Since our values are within 0.03 or less of Chefer et al. (2021), there may exist some hardware-based variance. Testing the segmentation task without PGD attack, we have found out that GradCAM still reports higher values than in Chefer et al. (2021) (details in Appendix C). We are not, however, able to account for Hu et al. (2024)'s significantly higher results.

Our reproduction of the results found in the study by Hu et al. (2024) had several limitations. Namely, our reproductions were limited to ImageNet datasets, due to us not having access to pre-trained models nor the exact methodology of approaching segmentation and perturbation tasks on COCO and CityScape datasets. Furthermore, our reproductions and extensions partially relied on a sparse code repository created by one author of Hu et al. (2024); unfortunately, most experiment scripts were missing, resulting in us making necessary assumptions. A more thorough discussion can be found in Section 7.1. We believe these limitations could be addressed by further research and more transparency from the original authors.

We will now go into more detail regarding the reproduction of Hu et al. (2024)'s experiments.

## 7.1 Reproducibility Limitations

We found it to be unproblematic to access the baseline methods' implementations discussed in Chefer et al. (2021) as well as AR (Xu et al., 2022) and incorporate it alongside DDS in both segmentation and classification tasks. Additionally, installation of the working environment (on a Linux-based workspace) proved to be fairly straightforward, making the reproduction of the qualitative demo very easy.

However, we encountered multiple hardships in reproducing the results from Hu et al. (2024). Firstly, an official repository with the code was not listed in the article at the time of writing and we had to base our implementations on a repository by one of the authors containing only a Jupyter notebook for visualizations. We later learned that this repository contained outdated implementations. Secondly, we found multiple discrepancies between the demo notebook and the information specified in the article. One example of this is a demo notebook varying the noise amount between cells, while the article specified a specific number.

As we lacked a full implementation of the experimental setup, we also found the information provided in the article to be incomplete, and we had to resort to assumptions made based on the limited implementations in the demo. This includes reporting classification accuracy as part of the results in Table 1 (the equivalent of Table 1 in this paper), which is never defined in the methodology, hence why it is omitted in our work. In addition, the number of input samples to be processed with DDS and subsequently combined was not precisely defined, leading us to specify the value of 2 for the two tasks taken from Hu et al. (2024)'s demo. This has an impact on the computational cost and energy consumed. Nonetheless, the conclusions regarding the environmental impact should still stand with a single pass-through of DDS.

Furthermore, some methods (taken from Chefer et al. (2021) and later used by Hu et al. (2024) and Xu et al. (2022)) were not set up in a computationally efficient manner: for example, the diffusion denoising process was set up to support a batch size of 1, which resulted in the under-utilization of GPUs and longer runtimes.

Lastly, the experiment specifications of the COCO and Cityscape datasets, the details of the segmentation and classification tasks, and how the corresponding ViTs are trained are lacking. This led us to focus on the ImageNet tasks exclusively.

## 7.2 Future Directions

A key future direction for DDS lies in improving its computational efficiency. To improve efficiency without compromising robustness, one promising approach is to adopt lighter-weight denoisers. Since the robustness of DDS stems from the consistency of denoising rather than model scale, it is feasible to replace large diffusion backbones with efficient, distilled denoising networks (Meng et al., 2023), which approximate the behaviour of guided diffusion models while enabling faster inference. These student models can be derived via score distillation or adversarial fine-tuning, and architectures such as compact U-Nets may serve as

appropriate candidates. Such substitutions can substantially reduce computational demands while preserving interpretability and robustness, making DDS more viable for real-time settings.

Additionally, DDS can benefit from reducing the number of diffusion steps through adaptive noise scheduling (Sahoo et al., 2024). While traditional diffusion models require hundreds of denoising iterations, recent work shows strong performance with as few as 3 to 5 steps. By leveraging dynamic step allocation or distilling guided diffusion models, DDS can preserve its effectiveness while significantly lowering computational cost.

### 7.3 Communication With Original Authors

We contacted the authors by email at the end of January, mostly regarding the implementation details. The authors made us aware that the code base that was published, which we have used to reproduce the results, is an outdated version and that the paper version will be published soon. However, the paper version was never made available to us. We contacted the authors again in April, once again requesting access to the updated codebase and requesting a meeting to discuss hyperparameter setup. At the time of writing, the authors have still not responded.

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

# A   Computational Resources

| Experiment | Model | Method | DDS | Wall Time (hh:mm) | CPU Util. (hh:mm) | CPU Eff. (%) |
|---|---|---|---|---|---|---|
| Segmentation | ViT | Raw Attention | × | 00:59 | 04:45 | 53.38 |
| | | | ✓ | 10:06 | 13:50 | 15.21 |
| | | Rollout | × | 00:57 | 04:43 | 55.53 |
| | | | ✓ | 10:04 | 15:41 | 17.31 |
| | | GradCAM | × | 00:48 | 04:35 | 62.68 |
| | | | ✓ | 09:46 | 13:30 | 15.35 |
| | | LRP | × | 00:57 | 04:43 | 55.46 |
| | | | ✓ | 10:04 | 13:48 | 15.22 |
| | | TA | × | 00:57 | 04:43 | 55.53 |
| | | | ✓ | 10:08 | 15:45 | 17.26 |
| | | AR | × | 00:58 | 04:51 | 55.40 |
| | | | ✓ | 10:06 | 15:44 | 17.30 |
| | DeiT | Raw Attention | × | 00:57 | 04:43 | 54.97 |
| | | | ✓ | 10:13 | 14:01 | 15.26 |
| | | Rollout | × | 00:57 | 04:43 | 55.13 |
| | | | ✓ | 10:03 | 15:40 | 17.32 |
| | | GradCAM | × | 00:43 | 04:29 | 69.57 |
| | | | ✓ | 09:53 | 13:36 | 15.31 |
| | | LRP | × | 00:57 | 04:42 | 55.43 |
| | | | ✓ | 10:03 | 13:46 | 15.23 |
| | | TA | × | 00:58 | 04:43 | 54.45 |
| | | | ✓ | 10:08 | 15:52 | 17.40 |
| | | AR | × | 00:56 | 04:43 | 55.94 |
| | | | ✓ | 10:08 | 15:45 | 17.28 |
| Classification[8] | ViT | Raw Attention | × | 00:38 | 05:33 | NA |
| | | Rollout | × | 00:37 | 05:39 | NA |
| | | TA | × | 00:37 | 05:26 | NA |
| | | | ✓ | 09:14 | 83:12 | NA |
| | | AR | × | 00:35 | 05:14 | NA |
| | | | ✓ | 09:07 | 83:16 | NA |

Table 4: Computational resources used in the image segmentation and classification tasks.

---

[8]Classification results are reported for negative perturbation and a PGD attack radius of $\frac{8}{255}$, for both generation of visualizations and subsequent classification. Runtimes for the full experiment are roughly 10 times longer. CPU Efficiency was not reported due to a small technical error on the server.

# B  Proofs of Theorems in Section 5

**Conjecture B.1** (Theorem 4.4 in Hu et al. (2024))**.** *If a function is an $(R, D_\alpha, \gamma, \beta, k, \|\cdot\|)$-faithful attention module for ViTs then if*

$$\gamma \leqslant -\log\left((1 - p_{(1)} - p_{(2)} + 2\left(\frac{1}{2}(p_{(1)}^{1-\alpha} + p_{(2)}^{1-\alpha})\right)^{\frac{1}{1-\alpha}}\right),$$

*we have for all $x'$ such that where $\|x - x'\| \leqslant R$,*

$$\arg\max_{g \in \mathcal{G}} \mathbb{P}(\bar{y}(x) = g) = \arg\max_{g \in \mathcal{G}} \mathbb{P}(\bar{y}(x') = g)$$

*where $\mathcal{G}$ is the set of classes, $p_{(i)}$ are $i$-th largest probabilities in $\{p_j\}$, where $p_j$ is the probability that $\bar{y}(x)$ returns the $j$-th class.*

**Conjecture B.2** (Theorem 5.1 in Hu et al. (2024))**.** *Consider the function $\tilde{w}$ where $\tilde{w} = Z(T(x + z))$, with $Z$ being the self attention module, $T$ a denoised diffusion model, $x$ input data and $z \sim \mathcal{N}(0, \sigma^2 I_{q \times n})$. Then it is an $(R, D_\alpha, \gamma, \beta, k, \|\cdot\|)$-faithful attention module for ViTs for any $\alpha > 1$ if for any input data $x$ we have*

$$\sigma^2 \leqslant \max\left\{\frac{\alpha R^2}{2\left(\frac{\alpha}{\alpha-1}\log\left(2k_0\left(\sum_{i \in \mathcal{S}} \tilde{w}_{i*}^\alpha\right)^{\frac{1}{\alpha}} + (2k_0)^{\frac{1}{\alpha}} \sum_{i \notin \mathcal{S}} \tilde{w}_{i*}\right) - \frac{1}{\alpha-1}\log(2k_0)\right)}, \frac{\alpha R^2}{2\gamma}\right\}$$

*where $k_0 = \lfloor(1 - \beta)k\rfloor + 1$ and $\mathcal{S}$ the set of last $k_0$ components in top-k indices and the top $k_0$ components out of top-k indices.*

## B.1  Theorem 5.1

*Proof.* Let $f$ be an $(R, D_\alpha, \gamma, \beta, k, \|\cdot\|)$-faithful attention module and

$$\gamma < -\log\left((1 - p_{(1)} - p_{(2)} + 2\left(\frac{1}{2}(p_{(1)}^{1-\alpha} + p_{(2)}^{1-\alpha})\right)^{\frac{1}{1-\alpha}}\right). \tag{1}$$

By Li et al. (2019) we have

**Lemma B.1** (Lemma 1 in Li et al. (2019))**.** *Let $P$ and $Q$ be two multinomial distributions over the same index set $\mathcal{I} = \{1, \ldots, k\}$. If the indices of the largest probabilies do not match on $P$ and $Q$, that is*

$$\arg\max_{i \in \mathcal{I}} p_i \neq \arg\max_{i \in \mathcal{I}} q_i$$

*then*

$$D_\alpha(P\|Q) \geqslant -\log\left((1 - p_{(1)} - p_{(2)} + 2\left(\frac{1}{2}(p_{(1)}^{1-\alpha} + p_{(2)}^{1-\alpha})\right)^{\frac{1}{1-\alpha}}\right)$$

*where $p_{(1)}, p_{(2)}$ are the two largest probabilities in the distribution $P$.*

Since $f$ is an $(R, D_\alpha, \gamma, \beta, k, \|\cdot\|)$-faithful attention module, it fulfils prediction robustness, thus

$$D_\alpha(\bar{y}(x)\|\bar{y}(x')) \leqslant \gamma$$

for any $x'$ such that $\|x - x'\| \leqslant R$. Therefore we have

$$D_\alpha(P\|Q) \leqslant \gamma < -\log\left((1 - p_{(1)} - p_{(2)} + 2\left(\frac{1}{2}(p_{(1)}^{1-\alpha} + p_{(2)}^{1-\alpha})\right)^{\frac{1}{1-\alpha}}\right)$$

based on equation 1 and $P = \bar{y}(x)$ and $Q = \bar{y}(x')$. Therefore we can conclude that

$$\arg\max_{i \in \mathcal{I}} p_i = \arg\max_{i \in \mathcal{I}} q_i$$

which is equivalent to

$$\arg\max_{g \in \mathcal{G}} \mathbb{P}(\bar{y}(x) = g) = \arg\max_{g \in \mathcal{G}} \mathbb{P}(\bar{y}(x') = g)$$

under conditions specified in Theorem 5.1. $\qquad\square$

### B.1.1 Counterexample

If the inequality in the equation 1 would be non-strict as it is in Conjecture B.1, we would not be able to reach this conclusion as in the equality case we would not be contradicting the conclusion of Lemma B.1. In fact we would have

$$D_\alpha(P\|Q) \leqslant \gamma \leqslant -\log\left((1 - p_{(1)} - p_{(2)} + 2\left(\frac{1}{2}(p_{(1)}^{1-\alpha} + p_{(2)}^{1-\alpha})\right)^{\frac{1}{1-\alpha}}\right)$$

and if we were to have a case where

$$\gamma = -\log\left((1 - p_{(1)} - p_{(2)} + 2\left(\frac{1}{2}(p_{(1)}^{1-\alpha} + p_{(2)}^{1-\alpha})\right)^{\frac{1}{1-\alpha}}\right)$$

then

$$D_\alpha(P\|Q) \leqslant -\log\left((1 - p_{(1)} - p_{(2)} + 2\left(\frac{1}{2}(p_{(1)}^{1-\alpha} + p_{(2)}^{1-\alpha})\right)^{\frac{1}{1-\alpha}}\right)$$

and thus by Lemma B.1

$$\arg\max_{g \in \mathcal{G}} \mathbb{P}(\bar{y}(x) = g) \neq \arg\max_{g \in \mathcal{G}} \mathbb{P}(\bar{y}(x') = g).$$

Alternatively, the proposed change to Conjecture B.1 could be avoided if in Definition 5.1 prediction robustness was specified as a strict inequality.

### B.2 Theorem 5.2

*Proof.* Let us consider the Rényi divergence of a DDS applied ViT $\tilde{w}$ for two inputs $x, x'$. Utilising the post-processing property of the Rényi divergence (Mironov, 2017) we can conclude

$$D_\alpha(\tilde{w}(x), \tilde{w}(x')) = D_\alpha(Z(T(x + z)), Z(T(x' + z)) \leqslant D_\alpha(x + z, x' + z)$$

and as $x + z \sim \mathcal{N}(x, \sigma^2 I)$, $x' + z \sim \mathcal{N}(x', \sigma^2 I)$

$$D_\alpha(\tilde{w}(x), \tilde{w}(x')) \leqslant \frac{\alpha\|x - x'\|}{2\sigma^2} \leqslant \frac{\alpha R}{2\sigma^2} \tag{2}$$

Having obtained a bound for Rényi divergence we can now proceed to proving the conditions for faithfulness: top-k robustness and prediction robustness.

If $D_\alpha(\tilde{w}(x), \tilde{w}(x')) < -\log\left((1 - p_{(1)} - p_{(2)} + 2\left(\frac{1}{2}(p_{(1)}^{1-\alpha} + p_{(2)}^{1-\alpha})\right)^{\frac{1}{1-\alpha}}\right)$ we have prediction robustness by contradiction of Lemma B.1 (see Appendix A.1). Thus if

$$\frac{\alpha R}{2\sigma^2} < -\log\left((1 - p_{(1)} - p_{(2)} + 2\left(\frac{1}{2}(p_{(1)}^{1-\alpha} + p_{(2)}^{1-\alpha})\right)^{\frac{1}{1-\alpha}}\right)$$

or equivalently

$$\sigma^2 > \frac{\alpha R^2}{-2\log\left((1 - p_{(1)} - p_{(2)} + 2\left(\frac{1}{2}(p_{(1)}^{1-\alpha} + p_{(2)}^{1-\alpha})\right)^{\frac{1}{1-\alpha}}\right)} \tag{3}$$

we fulfil the prediction robustness criterion.

Moving onto top-k robustness criterion we follow Hu et al. (2024) in introducing the lemma below.

**Lemma B.2** (Theorem 2 in Liu et al. (2021)). *Given two probability distributions $\tilde{w}$, $q$, the minimum change in Rényi divergence to violate the top-K robustness is*

$$\min_{q, V_k(\tilde{w}, q) \leqslant \beta} D_\alpha(\tilde{w} \| q) = -\log \left( 2k_0 \left( \frac{1}{2k_0} \sum_{i \in \mathcal{S}} \tilde{w}_{i^*}^{1-\alpha} \right)^{\frac{1}{1-\alpha}} + \sum_{i \notin \mathcal{S}} \tilde{w}_{i^*} \right)$$

Thus by Lemma B.2 if we have $\frac{\alpha R}{2\sigma^2} < -\log \left( 2k_0 \left( \frac{1}{2k_0} \sum_{i \in \mathcal{S}} \tilde{w}_{i^*}^{1-\alpha} \right)^{\frac{1}{1-\alpha}} + \sum_{i \notin \mathcal{S}} \tilde{w}_{i^*} \right)$ and equivalently

$$\sigma^2 > \frac{\alpha R^2}{-2\log \left( 2k_0 \left( \frac{1}{2k_0} \sum_{i \in \mathcal{S}} \tilde{w}_{i^*}^{1-\alpha} \right)^{\frac{1}{1-\alpha}} + \sum_{i \notin \mathcal{S}} \tilde{w}_{i^*} \right)} \tag{4}$$

top-K robustness will be guaranteed.

Putting equation 3 and equation 4 together we get

$$\sigma^2 \leqslant \max \left\{ \frac{\alpha R^2}{-2\log \left( 2k_0 \left( \frac{1}{2k_0} \sum_{i \in \mathcal{S}} \tilde{w}_{i^*}^{1-\alpha} \right)^{\frac{1}{1-\alpha}} + \sum_{i \notin \mathcal{S}} \tilde{w}_{i^*} \right)}, \right.$$
$$\left. \frac{\alpha R^2}{-2\log \left( 1 - p_{(1)} - p_{(2)} + 2 \left( \frac{1}{2} \left( p_{(1)}^{1-\alpha} + p_{(2)}^{1-\alpha} \right) \right)^{\frac{1}{1-\alpha}} \right)} \right\}.$$

$\square$

## C   Discrepancies of Results

We have run the segmentation task without attack on the baseline methods to compare the results with the original results of Chefer et al. (2021). The results are presented in Table 5. Raw Attention and TA match nearly perfectly. There is a slight difference between the scores for GradCAM, although the implementation uses the same architecture. The differences for LRP and Rollout stem from a slightly different implementation between the work of Chefer et al. (2021) and Hu et al. (2024)'s demo notebook, where we chose to follow the implementation of the latter for all of the main experiments. They differ slightly in the implementation of relevance propagation in the linear module. After adapting the architectural choices of baseline methods to those of Chefer et al. (2021), we were able to closely match results presented in the original paper, as shown in Table 5.

| Method | Pix. Acc | | | mAP | | | mIoU | | |
|---|---|---|---|---|---|---|---|---|---|
| | Chefer et al. | Ours - Chefer et al. (2021) | Ours - Hu et al. (2024) | Chefer et al. | Ours - Chefer et al. (2021) | Ours - Hu et al. (2024) | Chefer et al. | Ours - Chefer et al. (2021) | Ours - Hu et al. (2024) |
| Raw Attention | 67.84 | 67.87 | 67.87 | 80.24 | 80.24 | 80.24 | 46.37 | 46.37 | 46.37 |
| Rollout | 73.54 | 73.54 | 77.64 | 84.76 | 84.76 | 85.65 | 55.42 | 55.42 | 59.86 |
| GradCAM | 64.44 | 65.91 | 65.91 | 71.60 | 71.60 | 71.60 | 40.82 | 41.31 | 41.31 |
| LRP | 51.09 | 50.79 | 66.50 | 55.68 | 55.75 | 64.21 | 32.89 | 32.73 | 40.65 |
| TA | 79.70 | 79.72 | 79.72 | 86.03 | 86.02 | 86.02 | 61.95 | 61.99 | 61.99 |

Table 5: Comparison of interpretability methods on the segmentation task without an attack between the results of Chefer et al. (2021), our implementation following Chefer et al. (2021), and our implementation following Hu et al. (2024).

With no responses from Hu et al., we are left to speculate why we were not able to exactly reproduce their results. In this section, we address the discrepancies and show why our results are nonetheless valuable.

The full results for the segmentation task as found by Hu et al. (2024) are shown in Table 6. To more easily discuss the relative values within the table, we will present TA + DDS for ViT as a baseline to which the other values are compared. Such a conversion results in Table 7. The general patterns observed are as follows: DeiT metric is 0.01 higher than the corresponding metric for ViT and any Swin metric is 0.02 higher than the corresponding metric for ViT. Given a metric, model, and dataset, TA is 0.03 lower than TA with DDS. Then, in the order of TA, LRP, GradCAM, Rollout, and Raw Attention, each method is 0.02 lower than the previous one, except for Cla. Acc. where they are instead 0.01 lower. This pattern applies to all but two cells, namely DeiT TA + DDS on the COCO dataset.

The results of Hu et al. (2024) draw a clear image: the baseline methods in terms of performance rank in the order of: TA, LRP, GradCAM, Rollout, Raw Attention. However, for ViT, we have achieved an order of TA, Rollout, GradCAM, LRP, and Raw Attention for an attacked segmentation. Both aforementioned orders are not fully consistent with the methods' performance without an attack, both in Chefer et al. (2021) and our implementations. However, in those non-attacked scenarios as well as our DDS results, Rollout performs only slightly worse than TA, which is not consistent with values reported by Hu et al. (2024).

| Model | Method | ImageNet | | | | Cityscape | | COCO | |
|---|---|---|---|---|---|---|---|---|---|
| | | Cla. Acc. | Pix. Acc. | mIoU | mAP | Pix. Acc. | mIoU | Pix. Acc. | mIoU |
| ViT | Raw Attention | 0.78 | 0.65 | 0.54 | 0.82 | 0.72 | 0.62 | 0.8 | 0.7 |
| | Rollout | 0.79 | 0.67 | 0.56 | 0.84 | 0.74 | 0.64 | 0.82 | 0.72 |
| | GradCAM | 0.8 | 0.69 | 0.58 | 0.86 | 0.76 | 0.66 | 0.84 | 0.74 |
| | LRP | 0.81 | 0.71 | 0.6 | 0.88 | 0.78 | 0.68 | 0.86 | 0.76 |
| | TA | 0.82 | 0.73 | 0.62 | 0.9 | 0.8 | 0.7 | 0.88 | 0.78 |
| | TA + DDS | **0.85** | **0.76** | **0.65** | **0.93** | **0.83** | **0.73** | **0.91** | **0.81** |
| DeiT | Raw Attention | 0.79 | 0.66 | 0.55 | 0.83 | 0.73 | 0.63 | 0.81 | 0.71 |
| | Rollout | 0.8 | 0.68 | 0.57 | 0.85 | 0.75 | 0.65 | 0.83 | 0.73 |
| | GradCAM | 0.81 | 0.7 | 0.59 | 0.87 | 0.77 | 0.67 | 0.85 | 0.75 |
| | LRP | 0.82 | 0.72 | 0.61 | 0.89 | 0.79 | 0.69 | 0.87 | 0.77 |
| | TA | 0.83 | 0.74 | 0.63 | 0.91 | 0.81 | 0.71 | 0.89 | 0.79 |
| | TA + DDS | **0.86** | **0.77** | **0.66** | **0.94** | **0.84** | **0.74** | **0.89** | **0.79** |
| Swin | Raw Attention | 0.8 | 0.67 | 0.56 | 0.84 | 0.74 | 0.64 | 0.82 | 0.72 |
| | Rollout | 0.81 | 0.69 | 0.58 | 0.86 | 0.76 | 0.66 | 0.84 | 0.74 |
| | GradCAM | 0.82 | 0.71 | 0.6 | 0.88 | 0.78 | 0.68 | 0.86 | 0.76 |
| | LRP | 0.83 | 0.73 | 0.62 | 0.9 | 0.8 | 0.7 | 0.88 | 0.78 |
| | TA | 0.84 | 0.75 | 0.64 | 0.92 | 0.82 | 0.72 | 0.9 | 0.8 |
| | TA + DDS | **0.87** | **0.78** | **0.67** | **0.95** | **0.85** | **0.75** | **0.93** | **0.83** |

Table 6: Comparison of different methods on ImageNet, Cityscape, and COCO datasets as presented by Hu et al. (2024).

| Model | Method | ImageNet | | | | Cityscape | | COCO | |
|---|---|---|---|---|---|---|---|---|---|
| | | Cla. Acc. | Pix. Acc. | mIoU | mAP | Pix. Acc. | mIoU | Pix. Acc. | mIoU |
| ViT | Raw Attention | $A - 0.07$ | $B - 0.11$ | $C - 0.11$ | $D - 0.11$ | $E - 0.11$ | $F - 0.11$ | $G - 0.11$ | $H - 0.11$ |
| | Rollout | $A - 0.06$ | $B - 0.09$ | $C - 0.09$ | $D - 0.09$ | $E - 0.09$ | $F - 0.09$ | $G - 0.09$ | $H - 0.09$ |
| | GradCAM | $A - 0.05$ | $B - 0.07$ | $C - 0.07$ | $D - 0.07$ | $E - 0.07$ | $F - 0.07$ | $G - 0.07$ | $H - 0.07$ |
| | LRP | $A - 0.04$ | $B - 0.05$ | $C - 0.05$ | $D - 0.05$ | $E - 0.05$ | $F - 0.05$ | $G - 0.05$ | $H - 0.05$ |
| | TA | $A - 0.03$ | $B - 0.03$ | $C - 0.03$ | $D - 0.03$ | $E - 0.03$ | $F - 0.03$ | $G - 0.03$ | $H - 0.03$ |
| | TA + DDS | $A$ | $B$ | $C$ | $D$ | $E$ | $F$ | $G$ | $H$ |
| DeiT | Raw Attention | $A - 0.06$ | $B - 0.10$ | $C - 0.10$ | $D - 0.10$ | $E - 0.10$ | $F - 0.10$ | $G - 0.10$ | $H - 0.10$ |
| | Rollout | $A - 0.05$ | $B - 0.08$ | $C - 0.08$ | $D - 0.08$ | $E - 0.08$ | $F - 0.08$ | $G - 0.08$ | $H - 0.08$ |
| | GradCAM | $A - 0.04$ | $B - 0.06$ | $C - 0.06$ | $D - 0.06$ | $E - 0.06$ | $F - 0.06$ | $G - 0.06$ | $H - 0.06$ |
| | LRP | $A - 0.03$ | $B - 0.04$ | $C - 0.04$ | $D - 0.04$ | $E - 0.04$ | $F - 0.04$ | $G - 0.04$ | $H - 0.04$ |
| | TA | $A - 0.02$ | $B - 0.02$ | $C - 0.02$ | $D - 0.02$ | $E - 0.02$ | $F - 0.02$ | $G - 0.02$ | $H - 0.02$ |
| | TA + DDS | $A + 0.01$ | $B + 0.01$ | $C + 0.01$ | $D + 0.01$ | $E + 0.01$ | $F + 0.01$ | **G - 0.02** | **H - 0.02** |
| Swin | Raw Attention | $A - 0.05$ | $B - 0.09$ | $C - 0.09$ | $D - 0.09$ | $E - 0.09$ | $F - 0.09$ | $G - 0.09$ | $H - 0.09$ |
| | Rollout | $A - 0.04$ | $B - 0.07$ | $C - 0.07$ | $D - 0.07$ | $E - 0.07$ | $F - 0.07$ | $G - 0.07$ | $H - 0.07$ |
| | GradCAM | $A - 0.03$ | $B - 0.05$ | $C - 0.05$ | $D - 0.05$ | $E - 0.05$ | $F - 0.05$ | $G - 0.05$ | $H - 0.05$ |
| | LRP | $A - 0.02$ | $B - 0.03$ | $C - 0.03$ | $D - 0.03$ | $E - 0.03$ | $F - 0.03$ | $G - 0.03$ | $H - 0.03$ |
| | TA | $A - 0.01$ | $B - 0.01$ | $C - 0.01$ | $D - 0.01$ | $E - 0.01$ | $F - 0.01$ | $G - 0.01$ | $H - 0.01$ |
| | TA + DDS | $A + 0.02$ | $B + 0.02$ | $C + 0.02$ | $D + 0.02$ | $E + 0.02$ | $F + 0.02$ | $G + 0.02$ | $H + 0.02$ |

Table 7: Comparison of different methods on ImageNet, Cityscape, and COCO datasets as presented by Hu et al. (2024) using relative values. In bold, cells not following the general pattern.

# D    Original Results

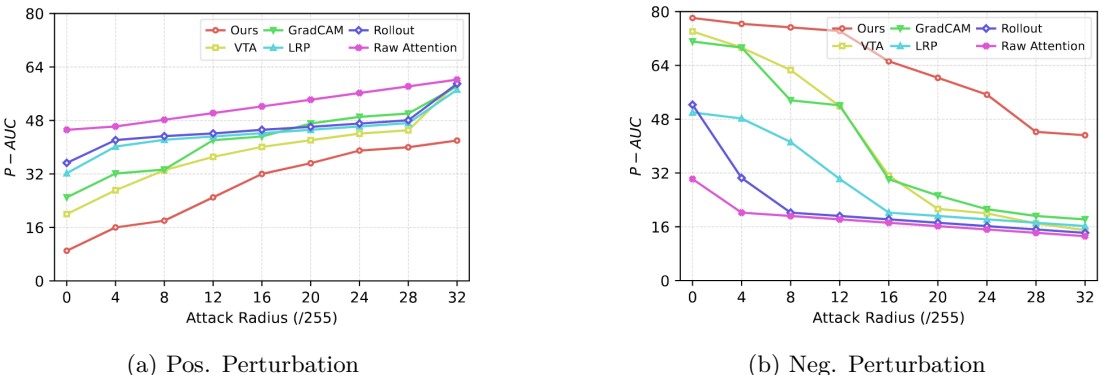

(a) Pos. Perturbation            (b) Neg. Perturbation

Figure 5: Taken from Hu et al. (2024). The AUC values of the interpretability methods in the image classification task per PGD attack radius under positive or negative perturbation. The AUC of the correctly classified images is calculated for all perturbation amounts: from 10% to 90% of the pixels removed in 10% increments. For positive perturbation, a lower AUC is better while for negative a higher AUC is better.

# E    Visualisations

Visualizations of all interpretability methods, with and without DDS, before and after perturbation, on three images. Noise level of $\frac{5}{255}$ was used both as the PGD attack radius and for DDS. The number of denoising steps was calculated according to the method provided in the original demo.

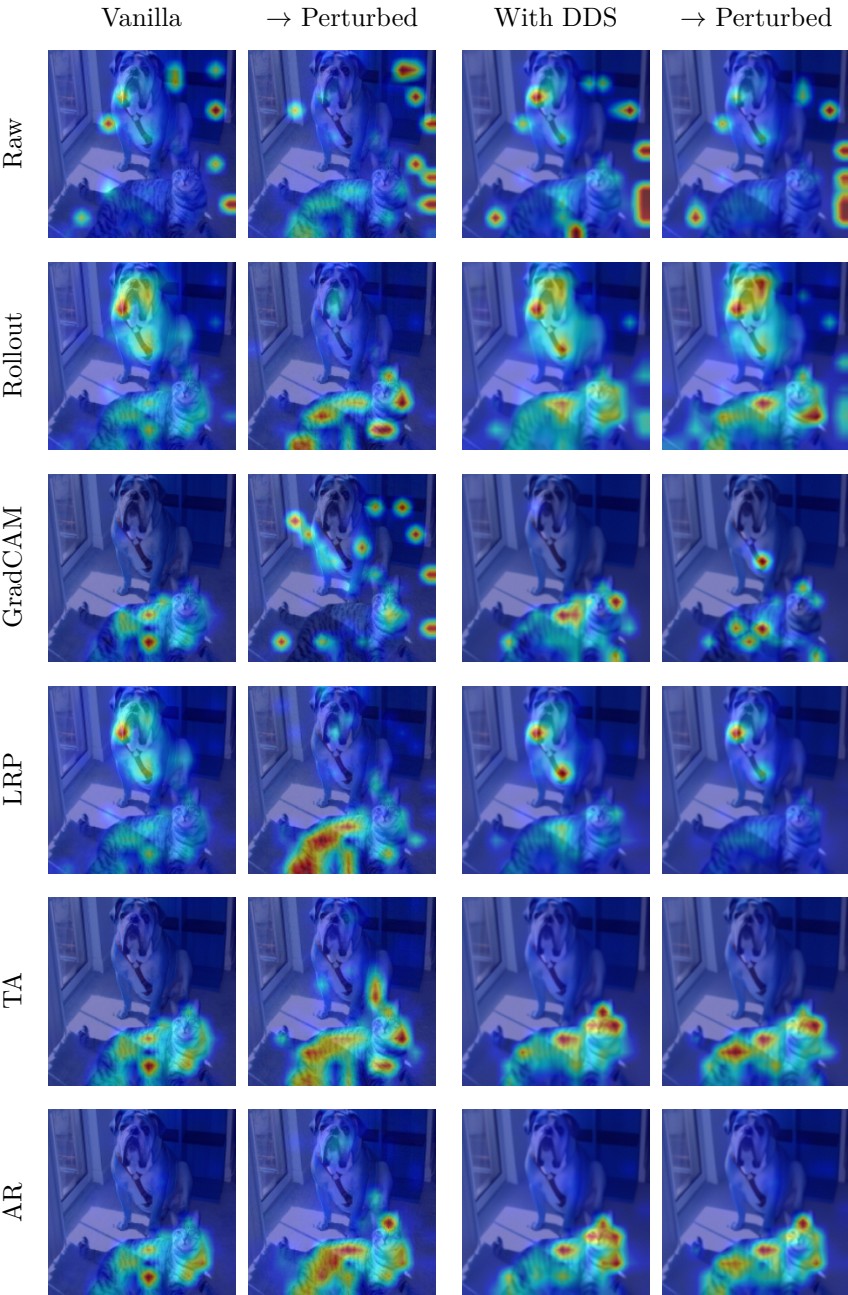

Figure 6: Visualizations on the cat and dog image. **Cat** as the predicted class.

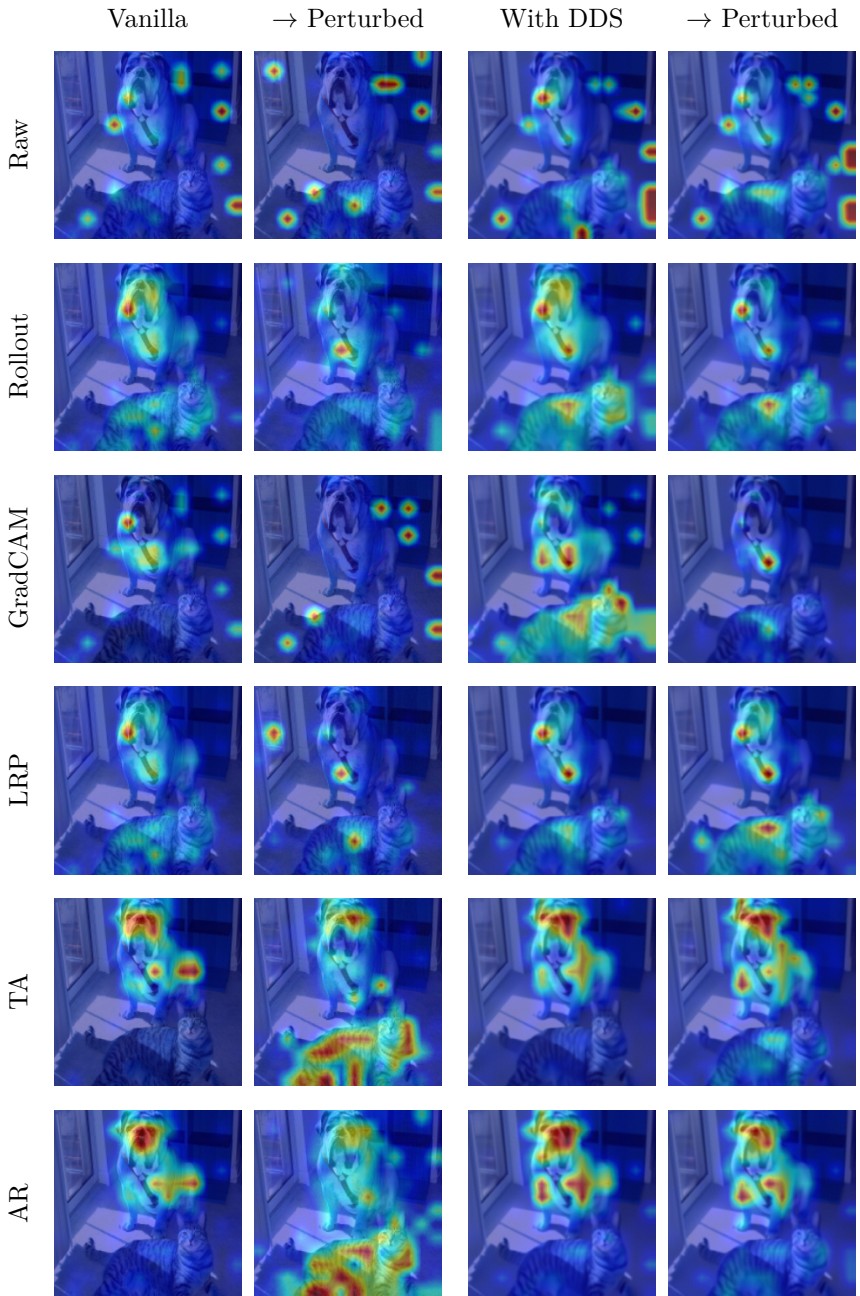

Figure 7: Visualizations on the cat and dog image. **Dog** as the predicted class.

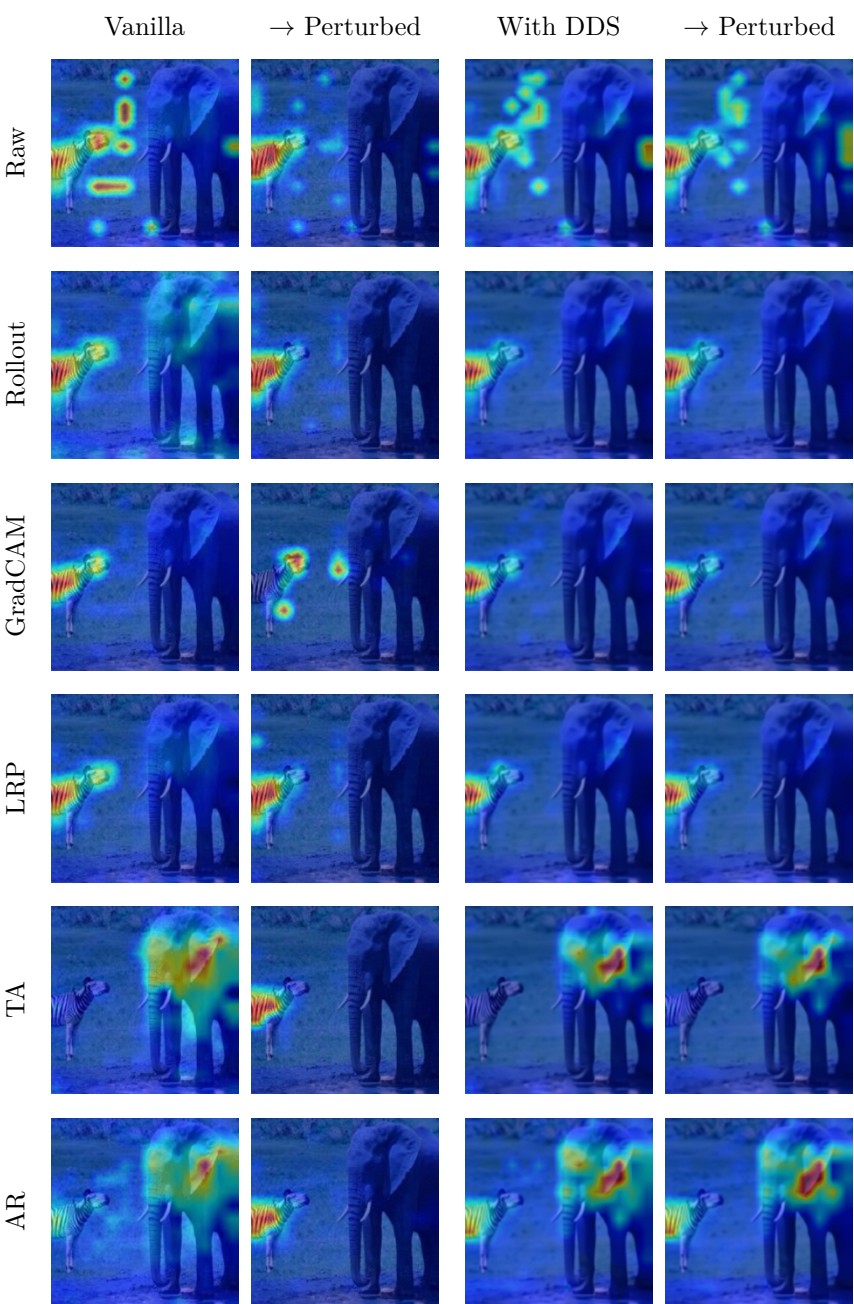

Figure 8: Visualizations on an elephant and zebra image. **Elephant** as the predicted class.

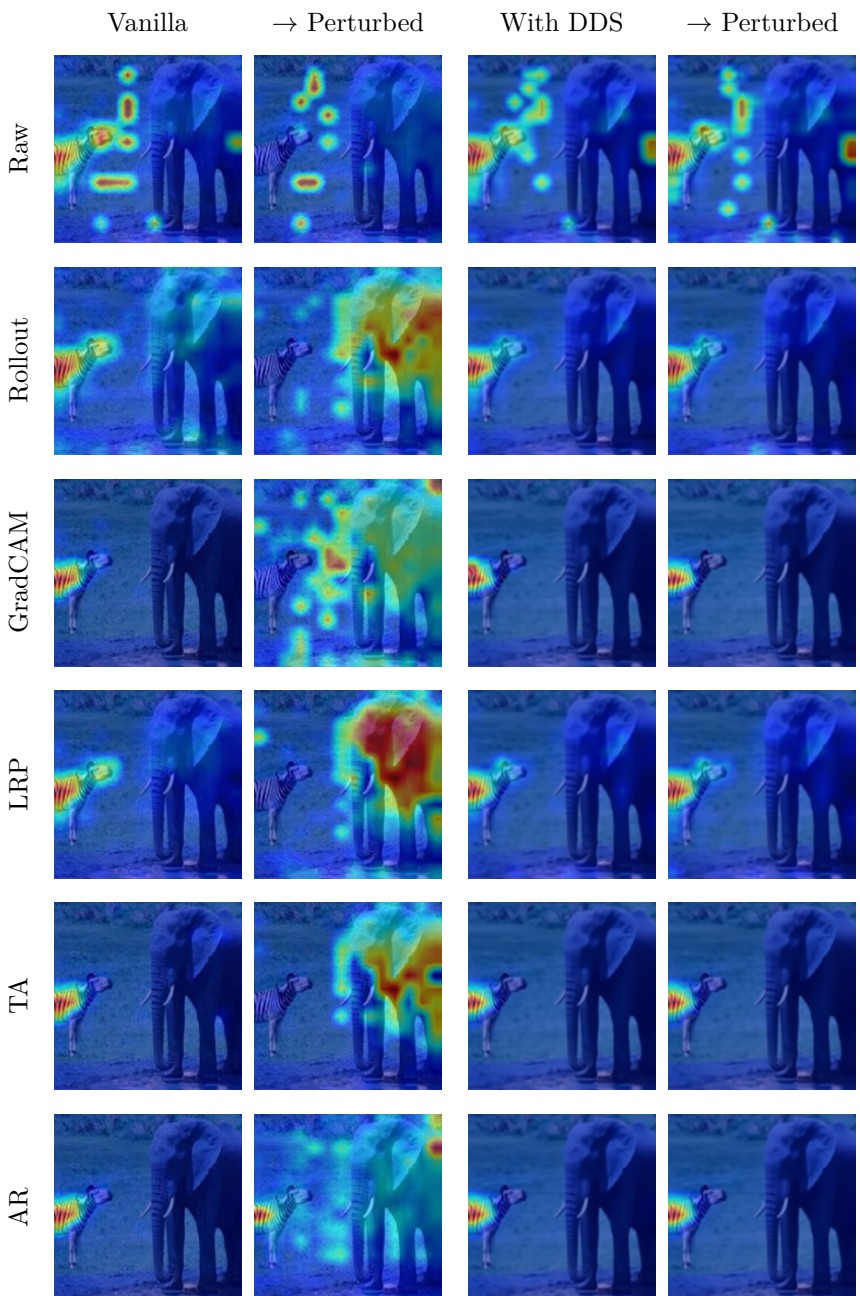

Figure 9: Visualizations on an elephant and zebra image. **Zebra** as the predicted class.

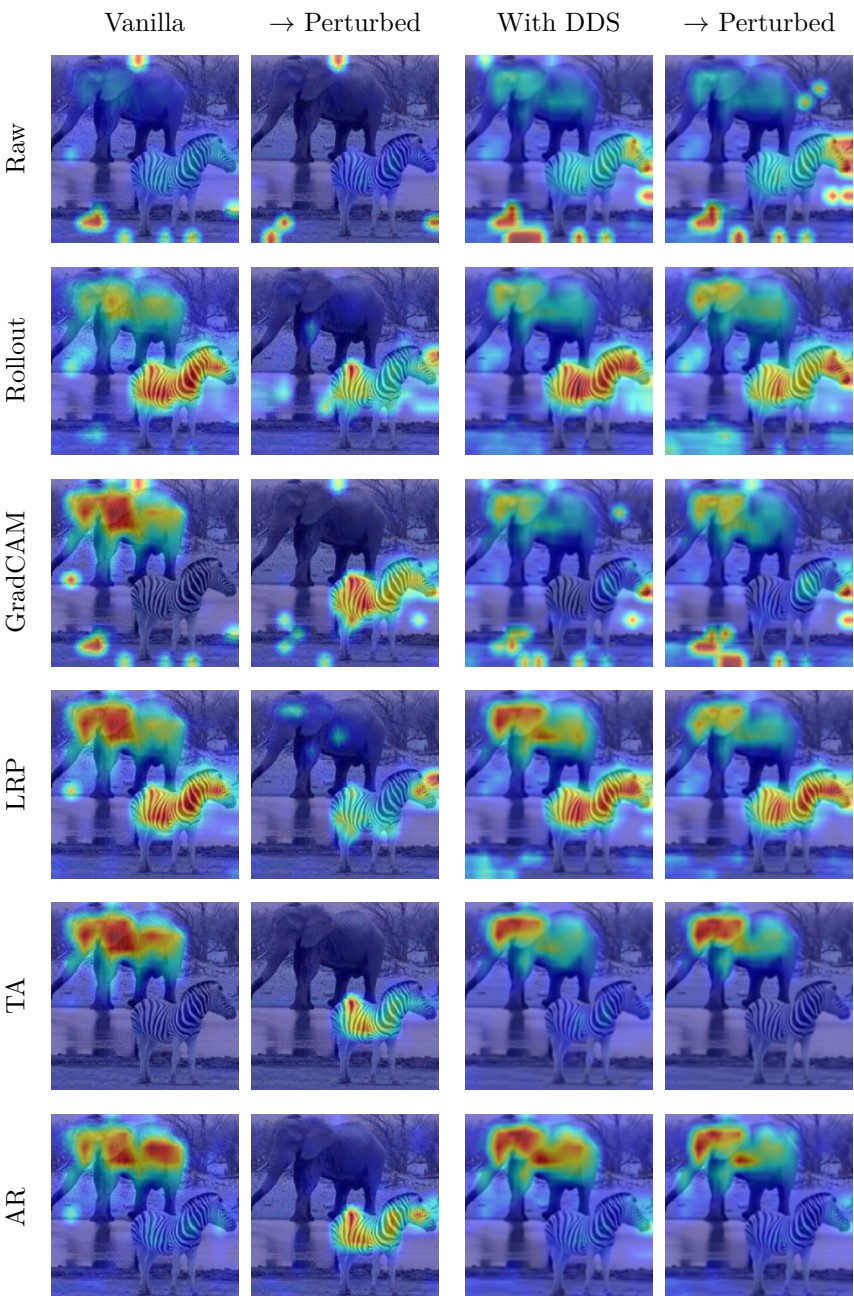

Figure 10: Visualizations on an elephant and zebra image. **Elephant** as the predicted class.

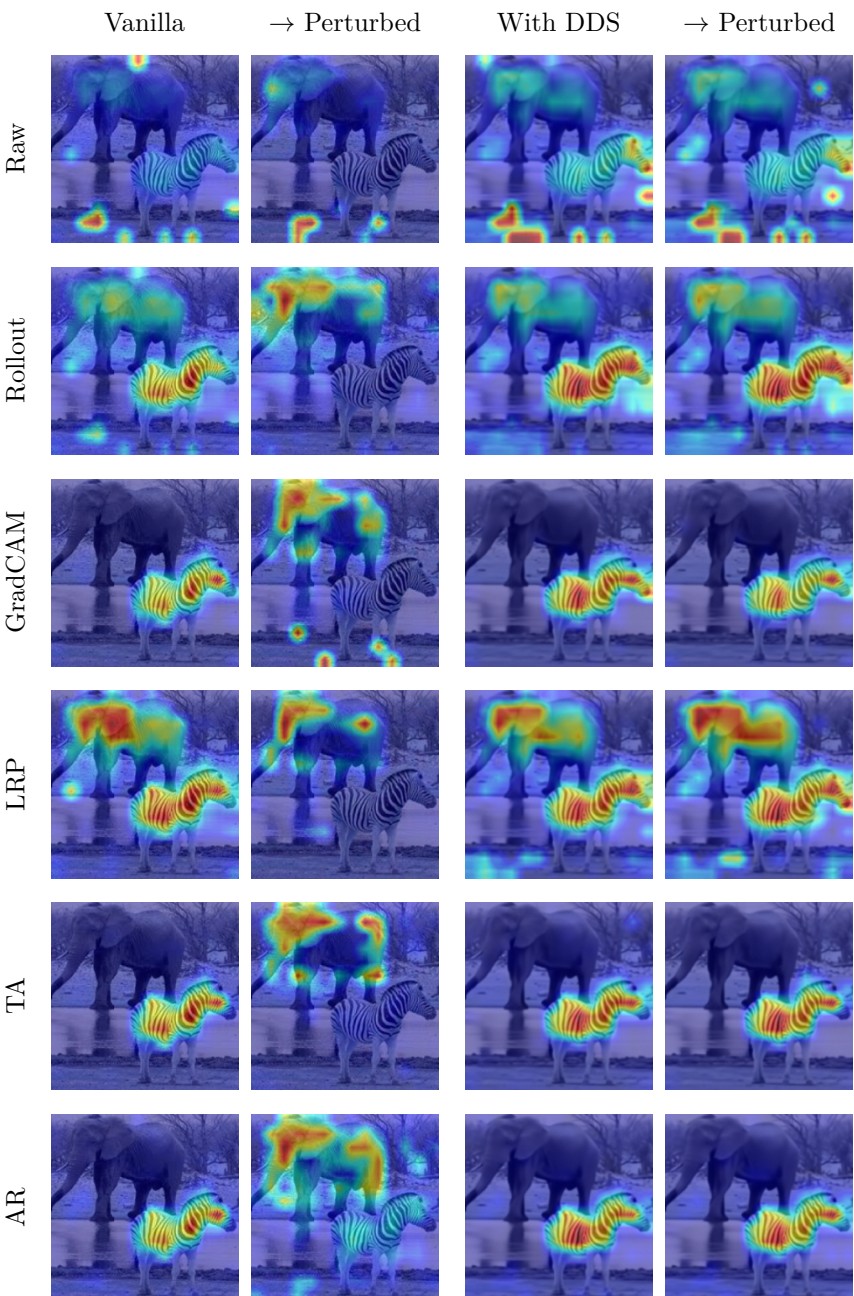

Figure 11: Visualizations on an elephant and zebra image. **Zebra** as the predicted class.

## F   Environmental Impact

| Model | Method | DDS | Execution Time (s) | Energy (J) | $CO_2$ Emitted (g) |
|-------|--------|-----|--------------------|------------|--------------------|
| ViT | Raw Attention | × | 3554 | 355400 | 37.53 |
|     |               | ✓ | 36266 | 6001885 | 616.86 |
|     | Rollout | × | 3390 | 561333 | 57.69 |
|     |         | ✓ | 36316 | 6560805 | 674.30 |
|     | GradCAM | × | 2912 | 296822 | 30.51 |
|     |         | ✓ | 36238 | 6376733 | 655.39 |
|     | LRP | × | 3385 | 533882 | 54.87 |
|     |     | ✓ | 35236 | 6220494 | 639.33 |
|     | TA | × | 3388 | 956766 | 98.33 |
|     |    | ✓ | 36494 | 4577928 | 470.51 |
|     | AR | × | 3488 | 430928 | 44.29 |
|     |    | ✓ | 36341 | 5316731 | 546.44 |
| DeiT | Raw Attention | × | 3417 | 516766 | 53.11 |
|      |               | ✓ | 36744 | 6324599 | 650.02 |
|      | Rollout | × | 3414 | 624185 | 64.15 |
|      |         | ✓ | 36154 | 6136458 | 630.68 |
|      | GradCAM | × | 2563 | 466887 | 47.98 |
|      |         | ✓ | 35545 | 6101154 | 627.06 |
|      | LRP | × | 3381 | 516766 | 53.11 |
|      |     | ✓ | 36149 | 6208825 | 638.12 |
|      | TA | × | 3457 | 399064 | 41.01 |
|      |    | ✓ | 36466 | 4530315 | 465.61 |
|      | AR | × | 3360 | 550719 | 56.60 |
|      |    | ✓ | 36449 | 6309426 | 648.46 |

Table 8: Environmental impact of one validation run per interpretability method and model in the segmentation task.

