# OpenReview forum: "[Re] Improving Interpretation Faithfulness for Vision Transformers"
_TMLR — Accepted by TMLR_

### Review · Reviewer_fy7y · 2025-03-11

**Summary Of Contributions:**

This paper reproduces the results of Faithful Vision Transformers (FViTs) proposed by Hu et al. (2024) alongside interpretability methods for Vision Transformers from Chefer et al. (2021) and Xu et al. (2022). Sufficient discussion and analysis are provided in this paper. Moreover, thorough experiments are conducted to show the reproduced results.

**Audience:**

Yes

**Claims And Evidence:**

Yes

**Requested Changes:**

Overall, I think this paper does a good job of re-viewing a published work. I suggest the authors carefully address my following concerns:
1. This paper solely conducts the experimental analysis, and there is no discussion about the correctness of the original mathematical proofs in Hu et al. (2024). I suggest the authors try to check their proofs and give us feedback on whether their proof is correct or not.
2. The discussion about the proposed extension is short. The authors should provide a detailed analysis of their extension with technical visualizations or algorithms to strengthen the contributions.
3. The reproduced results are worse than the original work. The authors should try to give some aspects to explore the potential reasons.
4. Visual analysis is limited. The authors should provide more interpretable visualization results to investigate the in-depth differences.

**Strengths And Weaknesses:**

Strengths
1. This paper provides a good review of previous work and tries to reproduce the results with additional novel designs.
2. The paper is well-organized and easy to follow.

Weaknesses
1. This paper solely conducts the experimental analysis, and there is no discussion about the correctness of the original mathematical proofs in Hu et al. (2024). I suggest the authors try to check their proofs and give us feedback on whether their proof is correct or not.
2. The discussion about the proposed extension is short. The authors should provide a detailed analysis of their extension with technical visualizations or algorithms to strengthen the contributions.
3. The reproduced results are worse than the original work. The authors should try to give some aspects to explore the potential reasons.
4. Visual analysis is limited. The authors should provide more interpretable visualization results to investigate the in-depth differences.

---

> ### Author Response · Authors · 2025-05-06
> **Detailed answers to requested changes**
>
> Thank you for your feedback. Below we include our detailed answers to changes requested by you.
>
> * **This paper solely conducts the experimental analysis, and there is no discussion about the correctness of the original mathematical proofs in Hu et al. (2024). I suggest the authors try to check their proofs and give us feedback on whether their proof is correct or not.**
>
> A new section "Mathematical Foundation" was added which includes discussion of the mathematical framework presented by Hu. Appendix A contains our proofs of the contents of section $5$. Note: not all theorems are discussed as the section is already quite lengthy and findings from the 2 theorems discussed can speak for the entirety of the mathematical framework presented by Hu.
>
> * **The discussion about the proposed extension is short. The authors should provide a detailed analysis of their extension with technical visualizations or algorithms to strengthen the contributions.**
>
> We have expanded the extensions also the classification task and have included a "Visualisations" section in the Appendix which shows the results of applying DDS to (new) methods.
>
> * **The reproduced results are worse than the original work. The authors should try to give some aspects to explore the potential reasons.**
>
> We included a subsection in the Discussion section (7.1) collecting all comments about this topic into one direct explanation.
>
> * **Visual analysis is limited. The authors should provide more interpretable visualization results to investigate the in-depth differences.**
>
> Connected to the earlier comment: there is now a Section in the Appendix with visualizations and two new figures were created: classification AUC values and CO2 emissions.

---

> > ### Comment · Reviewer_fy7y · 2025-05-15
> >
> > Thanks for the response, my concerns are addressed.

---

### Review · Reviewer_B8ER · 2025-03-21

**Summary Of Contributions:**

This paper aims to reproduce the results of Hu et al. (2024) to assess whether incorporating the Diffusion Denoised Smoothing (DDS) component into Vision Transformers (ViTs) enhances both interpretability and performance robustness of segmentation and classification tasks on the ImageNet dataset.

Beyond replication, the authors apply DDS to several interpretability methods—Raw Attention, Attention Rollout, GradCAM, Layerwise Relevance Propagation (LRP), Transformer Attribution (TA), and Attribution Rollout (AR)—to evaluate the general effectiveness of DDS for segmentation tasks. Additionally, they provide an analysis of the environmental impact of FViT (ViT augmented with DDS), contributing an important perspective that is often overlooked.

**Audience:**

No

**Broader Impact Concerns:**

No concerns on broader impact as this paper clearly analysed the environmental impact of their work.

**Claims And Evidence:**

Yes

**Requested Changes:**

1. Please consider expanding the background section to include brief, clear explanations of DDS and each interpretability method. The current introduction offers only a sentence per method, which may not be sufficient for readers unfamiliar with them.
2. The last paragraph of the introduction should focus more explicitly on the paper's contributions. In particular, clearly stating how your results match Hu et al. (2024)’s conclusion about DDS enhancing interpretability and robustness.
3. Why was PGD attack feasible for segmentation tasks but not for classification tasks? This seems counter-intuitive, as segmentation models typically are more computationally expensive.
4. Extending the experiments to include COCO, Cityscapes, and other models (e.g., Swin, DeiT) would significantly strengthen the work and more fully reproduce the original study’s scope.
5. Is the maximum perturbation in your PGD attack based on the $\ell_2$ or $\ell_\infty$ norm? It would be helpful to explicitly state this in the paper.
6. Do you know why your GradCAM results under attack outperform Chefer et al. (2021)’s original results without an attack? A brief analysis in the paper would be appreciated.
7. Can you elaborate on why your classification results differ from those in Hu et al. (2024)? Could the differences be due to using a small subset of images? Adding this discussion in the results section would help clarify the implications of your findings.

**Strengths And Weaknesses:**

### Strengths
1. The authors provide a thorough explanation of their implementation and openly share their code, which significantly contributes to reproducibility efforts within the community.
2. They replicate some of the results from Hu et al. (2024), generally confirming the original conclusion that DDS improves both performance and interpretability robustness for segmentation tasks. While some numerical results differ slightly, the overarching findings remain consistent.
3. The paper extends the application of DDS to additional interpretability methods, offering a broader evaluation of its robustness benefits.
4. The paper provides the environmental impacts of adding a DDS component which is very important and often neglected in publications.


### Weaknesses
1. The primary limitation of this work is the lack of comprehensiveness in the experimental results. The authors mentioned that is due to computational limitations or lack of openly available pretrained models or scripts/instructions for training models.That is understandable in general but for a reproducibility paper, it is crucial to have more comprehensive results. More complete replication would strengthen the contribution considerably Specifically:
    1. The paper only evaluates the ImageNet dataset, omitting COCO and Cityscapes datasets that were included in the original study. Expanding to these would provide a more complete validation.
     2. For classification attacks, the authors use only the positive/negative perturbation attack, while Hu et al. (2024) also included PGD attacks. The authors mention computational limitations, but this raises a question: if PGD attacks were feasible for segmentation (typically more resource-intensive), why not for classification?
     3. The study tests DDS only on Vanilla ViT, whereas the original paper evaluated it across Vanilla ViT, DeiT, and Swin ViT. Including more models would offer stronger generalization support.
2. Additionally, the presentation of the paper could be improved. For instance:
     1. The Introduction could provide a more detailed description of the paper’s contributions, especially the confirmation that their findings align with Hu et al. (2024) on segmentation task but not classification.
     2. The Background section would benefit from more thorough explanations of the interpretability methods (Attention Rollout, GradCAM, LRP, TA, AR) and DDS.
3. It seems like for classification tasks, the results do not match Hu et. al. (2024), but are more aligned with Chefer et al. (2021). A little more explanation on why there is such a discrepancy would be helpful.

---

> ### Author Response · Authors · 2025-05-06
> **Detailed answers to requested changes**
>
> Thank you for your feedback. Below we include our detailed answers to changes requested by you.
>
> * **Please consider expanding the background section to include brief, clear explanations of DDS and each interpretability method. The current introduction offers only a sentence per method, which may not be sufficient for readers unfamiliar with them.**
>
> The background section now includes concise and clear descriptions of DDS and each interpretability method used (Raw Attention, GradCAM, Rollout, LRP, Transformer Attribution), making it accessible to readers unfamiliar with these techniques.
>
> * **The last paragraph of the introduction should focus more explicitly on the paper's contributions. In particular, clearly stating how your results match Hu et al.’s conclusion about DDS enhancing interpretability and robustness.**
>
> The final paragraph of the introduction has been rewritten to explicitly outline our paper's contributions and how they relate to and support Hu et al.’s claims regarding DDS’s effect on interpretability and adversarial robustness.
>
> * **Why was PGD attack feasible for segmentation tasks but not for classification tasks? This seems counter-intuitive, as segmentation models typically are more computationally expensive.**
>
> We extended our experiments to include a PGD attack at different attack radiuses for classification under perturbation.
>
> * **Extending the experiments to include COCO, Cityscapes, and other models (e.g., Swin, DeiT) would significantly strengthen the work and more fully reproduce the original study’s scope.**
>
> We attempted to addresses this limitation as much as possible with our resources. We expanded the image segmentation task to include a second model (DeiT), have multiplied the number of images used in the classification task, and have added a PGD attack to the classification task. Unfortunately, due to lack of scripts and models for other datasets, we still focused only on ImageNet.
>
> * **Is the maximum perturbation in your PGD attack based on the $l_2$ or $l_\infty$ norm? It would be helpful to explicitly state this in the paper.**
>
> The threat model is $l_\infty$. We expanded the paper to mention this.
>
> * **Do you know why your GradCAM results under attack outperform Chefer et al. (2021)’s original results without an attack? A brief analysis in the paper would be appreciated.**
>
> We do not have a clear answer, but an attempt at an explanation is made in Discussion section now.
>
> * **Can you elaborate on why your classification results differ from those in Hu et al. (2024)? Could the differences be due to using a small subset of images? Adding this discussion in the results section would help clarify the implications of your findings.**
>
> We expanded the task with a PGD attack to classification under perturbation and significantly expanded the data subset to include 4000 images. The results now broadly agree with those of Hu et al..

---

> > ### Comment · Reviewer_B8ER · 2025-05-26
> >
> > Thanks for addressing my comments.

---

### Review · Reviewer_woXt · 2025-04-29

**Summary Of Contributions:**

This study undertakes a reproduction of Hu et al. (2024)’s Improving Interpretation Faithfulness for Vision Transformers and introduces two extensions. First, it applies Denoised Diffusion Smoothing (DDS) to Attribution Rollout method (Xu et al., 2022) alongside the originally tested Transformer Attribution, thereby broadening the scope of DDS’s applicability. Second, it systematically evaluates DDS as a plug‐and‐play enhancement across a suite of ViT interpretability techniques—Raw Attention, Rollout, GradCAM, LRP, TA, and AR. Additionally, this paper measures the additional computational cost and carbon footprint incurred by diffusion based smoothing.

Results show that DDS improves AR’s segmentation performance and provides gains for LRP and TA, while having minimal effect on Raw Attention, Rollout, and GradCAM. The cost analysis reveals that DDS substantially increases runtime and CO₂ emissions, suggesting a clear trade‐off between robustness and efficiency.

**Audience:**

No

**Broader Impact Concerns:**

No broader impact concerns were identified.

**Claims And Evidence:**

Yes

**Requested Changes:**

- A focused analysis of why DDS substantially enhances some methods (LRP, TA, AR) but has little effect on other techniques (Raw Attention, Rollout, GradCAM) would provide valuable insight into the underlying mechanisms governing its robustness.
- Investigate and propose concrete optimizations—such as lighter-weight denoising models—to improve the computational efficiency of DDS without compromising robustness.
- Reach out to the original authors to verify and align hyper-parameter settings and implementation details, ensuring the reproduced results closely match the published values and thus strengthen the study’s credibility.

**Strengths And Weaknesses:**

The authors design a broad evaluation that mirrors Hu et al.’s original experiments. By extending DDS to Attribution Rollout, they demonstrate that smoothing can enhance newer transformer‐based explanations, achieving the highest segmentation accuracy and IoU under attack. Finally, their inclusion of energy‐usage measurements and CO₂ estimates brings an often-neglected sustainability perspective to interpretability research, quantifying the roughly ten-fold increase in computational and environmental cost that practitioners should weigh.

However, several limitations remain. Many baseline metrics diverge from Hu et al.’s reported values—GradCAM’s mIoU and mAP under attack being notable examples—suggesting unresolved discrepancies in implementation or evaluation. The paper also highlights DDS’s low compute efficiency without proposing strategies to mitigate it. Moreover, beyond combining AR with DDS, it offers little methodological innovation or theoretical insight into why smoothing benefits some attribution methods but not others.

---

> ### Author Response · Authors · 2025-05-06
> **Detailed answers to requested changes**
>
> Thank you for your feedback. Below we include our detailed answers to changes requested by you.
>
> * **A focused analysis of why DDS substantially enhances some methods (LRP, TA, AR) but has little effect on other techniques (Raw Attention, Rollout, GradCAM) would provide valuable insight into the underlying mechanisms governing its robustness.**
>
> There was an implementation issue when evaluating DDS's impact on ViT interpretability, which has been corrected and now all methods improve under DDS, though some improvements are minor. DeiT interpretability is not uniformly improved. Both minor and no-improvement scenario now has a postulated explanation in the first paragraph of discussion.
>
> * **Investigate and propose concrete optimizations—such as lighter-weight denoising models—to improve the computational efficiency of DDS without compromising robustness.**
>
> In discussion we now propose potential directions for developing methods of decreasing computational load of FViTs. As our study developed in the direction of plug-in functionality of DDS, we deemed the issue of efficiency to be a good direction for future research.
>
> * **Reach out to the original authors to verify and align hyper-parameter settings and implementation details, ensuring the reproduced results closely match the published values and thus strengthen the study’s credibility.**
>
> Our history of communication with original authors is presented in the final subsection of the article. Our two requests for access to finalized codebase the original paper has been based on have not been accepted and so was our request for meeting to discuss experimental setup.

---

> ### Comment · Reviewer_woXt · 2025-05-29
>
> Thank you for your detailed response; my concerns have now been addressed

---

### Decision · Action_Editor_SGYd · 2025-05-31

**Recommendation:** Accept as is

**Audience:**

Yes

**Audience Explanation:**

In addition to replicating previous work, the authors extend earlier conclusions by applying DDS to several interpretability methods, such as Attention Rollout and Grad-CAM. The results indicate that DDS is effective as a plug-in method for improving interpretability robustness. Meanwhile, the experimental findings also challenge the claims in original paper regarding DDS’s efficiency on large-scale data, as evidenced by the significant runtime required. In response to requests from reviewers B8ER, woXt, and fy7y, a detailed discussion of the experimental results has been added to the rebuttal to help understanding. I agree with the consensus of all reviewers that this work could benefit some members of the TMLR audience, particularly those interested in interpretable machine learning for vision tasks.

**Claims And Evidence:**

Yes

**Claims Explanation:**

This paper focuses on reproducing the results of "Improving Interpretation Faithfulness for Vision Transformers". The major claims in this paper are: (1) denoised diffusion smoothing (DDS) improves interpretability robustness in both segmentation and classification tasks, and (2) extending previous findings, DDS combined with other interpretability methods can further enhance robustness under attack. Before the rebuttal, reviewers raised concerns about the comprehensiveness of the experiments across different datasets and models (B8ER), as well as the reproduction results being lower than those reported in the original paper (woXt, fy7y). The rebuttal addressed these concerns point by point, and after the rebuttal, all reviewers agreed that the experiments on classification and segmentation tasks, along with the visualization analysis, support the paper's claims.